# Optimal Transport with Symmetry Groups

**Jiechao Zhang**[1]   **Huichun Zhang**[2]   **Jian Sun**[1]   **Wei Zeng**[1]

## Abstract

We propose a novel algorithm that accelerates optimal transport by exploiting intrinsic symmetries induced by finite group actions. The core of our approach is to recover the orbit decomposition and the associated algebraic structure directly from the cost matrix—without requiring prior knowledge of the group—and to reduce the original transport problem to a substantially smaller problem on the orbit space. This reduction preserves optimality while achieving a significant drop in computational complexity. We develop efficient solvers for two central classes of optimal transport: linear OT and entropy-regularized OT. Experiments on synthetic data, real-world image datasets, and molecular graph data confirm the efficiency and robustness of the method. To our knowledge, this work is the first to systematically incorporate symmetry groups into optimal transport, providing both a theoretical framework and a practical pathway to computational acceleration.

## 1. Introduction

Discrete optimal transport (OT) computes an optimal coupling between two discrete probability distributions in a metric space, minimizing the total transport cost. It has become a fundamental tool in machine learning and data analysis, with applications ranging from image matching (Papadakis, 2015; Liu et al., 2020) and domain adaptation (Courty et al., 2016; Gu et al., 2022) to generative modeling (Salimans et al., 2018) and computational biology (Schiebinger et al., 2019). Two of the most widely used formulations are linear OT (LOT) (Kantorovich, 1942), which is solved as a linear program, and entropy-regularized OT (EROT) (Knight, 2008; Cuturi, 2013; Blondel et al., 2018; Peyré et al., 2019; Guo et al., 2020), which adds an entropic penalty for numer-

[1]School of Mathematics and Statistics, Xi'an Jiaotong University, Xi'an, China [2]School of Mathematics, Sun Yat-sen University, Guangzhou, China. Correspondence to: Wei Zeng <wz@xjtu.edu.cn>.

*Proceedings of the 43rd International Conference on Machine Learning*, Seoul, South Korea. PMLR 306, 2026. Copyright 2026 by the author(s).

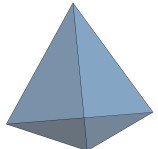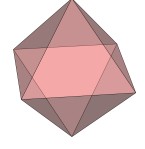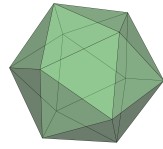

*(a)* Tetrahedron     *(b)* Octahedron     *(c)* Icosahedron

*Figure 1.* Point sets exhibiting non-cyclic rotational symmetries. For example, the tetrahedron *(a)* has 12 rotational symmetries forming a non-abelian group. These symmetries cannot be described by cyclic groups and thus motivate the general framework presented here.

ical stability and can be computed efficiently and scalably by Sinkhorn–Knopp algorithm(Cuturi, 2013).

To tackle the cubic complexity of exact OT, many works exploit structural properties of the data or the cost matrix, e.g., algorithms that utilize the low-rankness of the input data (Tenetov et al., 2018; Altschuler et al., 2019), several algorithms utilize the Gibbs kernel structure of the input cost matrix in the Sinkhorn algorithm (Getreuer, 2013; Solomon et al., 2015; Bonneel et al., 2016; Peyré et al., 2019). Recently, Takeda et al. (2024); Takeda & Akagi (2025) proposed an acceleration for data with cyclic symmetry, which induces a circulant block structure in the cost matrix and allows the problem to be reduced to a smaller scale. While effective, this approach is inherently limited to cyclic symmetries and critically depends on the points being presented in a specific cyclic ordering that exposes the block pattern. In practice, many datasets exhibit more complex, non-commutative symmetries (Miller, 1973; Armstrong, 1997; Butler, 2012)—such as the full rotation groups of polyhedra (Figure 1)—that cannot be described by simple cyclic arrangements. Manually identifying or imposing the correct block structure for such groups is often infeasible, limiting the applicability of existing symmetry-based accelerations.

This work introduces a general symmetry-aware optimal transport framework that overcomes these limitations. We consider OT problems where source and target point sets lie in the same metric space and share a common finite symmetry group. The key observation is that such symmetry introduces redundancy: the optimal transport plan can be shown to be group-invariant, allowing the problem to be

reduced to a lower-dimensional one defined on the orbit space. More importantly, we demonstrate that the orbit structure can be automatically recovered directly from the cost matrix, without any prior knowledge of the group or a predefined point ordering. This is possible under a mild and natural condition: points belonging to distinct orbits must produce distinct distance patterns, a property that holds for generic symmetric configurations.

The main contributions of this work are:

**General applicability to arbitrary finite group actions.** Unlike prior methods restricted to cyclic symmetries, our framework handles any finite symmetry group—including non-commutative groups such as the tetrahedral $T$, octahedral $O$, and icosahedral $I$ rotation groups. This enables acceleration for a much broader class of symmetric structures, overcoming a key limitation of earlier work.

**Automatic orbit identification from the cost matrix.** We provide a simple yet robust procedure to infer the orbit decomposition directly from pairwise distances, requiring no explicit block decomposition or special input ordering. The method is thus invariant to arbitrary permutations of the points, making it suitable for practical scenarios where data ordering is not controlled.

In parallel independent work, Wang et al. (2026) studied the existence of group-invariant optimal transport maps under general topological groups, extending the theoretical analysis beyond the finite group setting considered here. The present paper focuses on the algorithmic exploitation of finite group actions with automatic orbit identification and computational acceleration.

**Conflict of Interest Disclosure** The authors declare no competing interests.

## 2. Preliminaries

### 2.1. ROT: Regularized Optimal Transport

In classical optimal transport, the source and target domains often reside in the same metric space, with costs given by pairwise distances(Howard, 2003). To generalize this formulation, we define regularized optimal transport (ROT) in the general setting of metric spaces.

Let $(M, d)$ be a metric space. We consider two finite, ordered subsets as our source and target domains:

$$S = \{s_i\}_{i=1}^{|S|} \quad \text{and} \quad T = \{t_j\}_{j=1}^{|T|}.$$

On these sets, we define discrete probability measures

$$p = (p_i)_{i=1}^{|S|} \quad \text{and} \quad q = (q_j)_{j=1}^{|T|},$$

satisfying $\sum_{i=1}^{|S|} p_i = 1$ and $\sum_{j=1}^{|T|} q_j = 1$, respectively.

The cost matrix $C \in \mathbb{R}^{|S| \times |T|}$ is induced by the metric $d$, with entries $C_{i,j} = d(s_i, t_j)$. Within this setting, ROT is defined as

$$\min_{P \in \mathbb{R}_+^{|S| \times |T|}} \quad \langle C, P \rangle + \sum_{i=1}^{|S|} \sum_{j=1}^{|T|} \phi(P_{i,j}) \qquad (1)$$
$$\text{s.t.} \quad P\mathbf{1}_{|T|} = p, \quad P^\top \mathbf{1}_{|S|} = q,$$

where $\langle C, P \rangle = \sum_{i,j} C_{i,j} P_{i,j}$ denotes the Frobenius inner product, $\mathbf{1}_k$ is the all-ones vector in $\mathbb{R}^k$, and $\phi : \mathbb{R}_+ \to \mathbb{R}$ is a convex regularization function.

Equation (1) generalizes classical OT formulations. Notable examples include linear optimal transport (LOT), obtained with $\phi(x) \equiv 0$, and entropy-regularized OT (EROT), corresponding to $\phi(x) = \lambda x(\log x - 1)$.

### 2.2. Symmetry Groups and Group Actions

Formulating OT in a metric space allows symmetries to be treated geometrically. In this work, we model symmetries by the isometry groups under which the source and target sets are invariant.

Let $E$ be a subset of a metric space $(M, d)$. An isometry of $M$ that maps $E$ onto itself is called a symmetry of $E$. The set of all such symmetries,

$$G^E := \{ f \in \text{Isom}(M) \mid f(E) = E \},$$

forms a group under composition, called the symmetry group of $E$. $\text{Isom}(M)$ denotes the isometry group of $M$.

Once the symmetry group $G^E$ of the set $E$ is specified, $E$ can be partitioned into orbits under the group action. For any $x \in E$, its orbit is

$$\text{Orb}(x) := \{gx \mid g \in G^E\}.$$

The cardinality of an orbit depends on the geometric position of the point. Points located at positions with higher symmetry may have orbits smaller than the group order. To ensure a uniform analysis across all points, we restrict our attention to free group actions. That is, we assume

$$\text{Stab}(x) := \{g \in G^E \mid gx = x\} = \{e\},$$

where $e$ denotes the identity element. The classical orbit–stabilizer theorem (Burnside, 1911) guarantees that every orbit attains the maximal possible size:

$$|\text{Orb}(x)| = |G^E|, \quad \forall x \in E. \qquad (2)$$

Under free actions, all orbits have equal size, allowing a uniform orbit decomposition. This property is crucial for deriving the orbit-reduced formulation and complexity reduction. Concrete examples of symmetry groups, orbits and free actions are provided in Appendix A.1. For a general introduction to group theory, we refer to the standard textbook by Scott (2012).

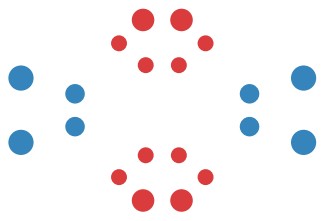

*Figure 2.* An illustration of SG-ROT in $\mathbb{R}^2$. Red and blue points represent the source and target domains, respectively. The size of each point corresponds to its measure. Both domains share the dihedral group $D_2$ as their symmetry group, with $m = 3$ source orbits and $n = 2$ target orbits. The group is generated by two mutually orthogonal reflections, and the probability measure is constant on each orbit.

## 3. SG-ROT: ROT with Symmetry Groups

This section studies regularized optimal transport in the presence of finite symmetry groups. We refer to this problem as regularized optimal transport with symmetry groups (SG-ROT).

### 3.1. Group Invariance of SG-ROT

We begin by stating the basic symmetry assumptions that define the SG-ROT setting.

**Assumption 3.1.** Let the source and target domains $S$ and $T$ share a common finite symmetry group $G$ (i.e., $G^S = G^T = G$), which acts freely on both domains. Furthermore, the probability measures $p$ on $S$ and $q$ on $T$ are constant on each group orbit: if two points belong to the same orbit, they are assigned the same measure.

A regularized optimal transport problem satisfying Assumption 3.1 will be called SG-ROT. Under this assumption, the group $G$ acts freely on both $S$ and $T$. Consequently, by (2), there exist positive integers $m$ and $n$ such that

$$|S| = m|G|, \quad |T| = n|G|, \tag{3}$$

where $m$ and $n$ count the numbers of orbits in the source and target domains, respectively. Figure 2 illustrates this symmetric setup.

The group action of $G$ on $S$ and $T$ permutes the point indices, thereby inducing corresponding permutations on the rows and columns of the cost matrix $C$. These permutations can be represented by permutation matrices $T_g^s \in \mathbb{R}^{m|G| \times m|G|}$ and $T_g^t \in \mathbb{R}^{n|G| \times n|G|}$, defined for each $g \in G$ by

$$(T_g^s)_{i,j} = \begin{cases} 1, & gs_j = s_i, \\ 0, & \text{otherwise}, \end{cases} \quad (T_g^t)_{i,j} = \begin{cases} 1, & gt_j = t_i, \\ 0, & \text{otherwise}. \end{cases}$$

Since each $g \in G$ is an isometry, we have $d(gs_i, gt_j) =$

$d(s_i, t_j)$, which is equivalent to the matrix relation

$$T_g^s C (T_g^t)^\top = C, \quad \forall g \in G. \tag{4}$$

Thus, $C$ is invariant under the action of $G$.

A key structural property of SG-ROT is that it always admits a group-invariant optimal solution. This is formalized in the following lemma, which is proved via a group-averaging argument (see Appendix A.2).

**Lemma 3.2.** *For SG-ROT, there exists an optimal transport plan $P$ satisfying*

$$T_g^s P (T_g^t)^\top = P, \quad \forall g \in G. \tag{5}$$

Imposing the invariance condition (5) on the original problem (1) and using the orbit-size relations (3), we obtain the following symmetry-constrained formulation of SG-ROT:

$$\min_{P \in \mathbb{R}_+^{m|G| \times n|G|}} \quad \langle C, P \rangle + \sum_{i=1}^{m|G|} \sum_{j=1}^{n|G|} \phi(P_{i,j})$$
$$\text{s.t.} \quad P\mathbf{1}_{n|G|} = p, \quad P^\top \mathbf{1}_{m|G|} = q, \tag{6}$$
$$T_g^s P (T_g^t)^\top = P, \quad \forall g \in G.$$

Equation (6) will serve as our working definition of SG-ROT; its equivalence to the original SG-ROT problem (i.e., (1) under Assumption 3.1) is proved in Appendix A.3.

### 3.2. OR-SG-ROT: Orbit-Reduced SG-ROT

Symmetry induces redundancy in the transport plan under group actions, which suggests that SG-ROT should be reducible to a lower-dimensional problem on the orbit space.

In practical settings, we often have access only to the pairwise distances encoded in the cost matrix $C$, without explicit knowledge of the underlying group. Therefore, to realize this reduction, we must first identify which points belong to the same orbit using only the information contained in $C$.

For a source point $s_i \in S$, we define its distance pattern to the target set as the multiset

$$\text{Pattern}(s_i) := \{d(s_i, t_j) \mid t_j \in T\}.$$

Patterns for target points are defined analogously. A fundamental observation is that orbit identification from $C$ is only possible if points from different orbits produce distinct patterns. If two distinct orbits generated identical multisets of distances to the other set, they would be fundamentally indistinguishable from $C$ alone; no method could recover the correct orbit structure from the distance information.

We therefore adopt the following condition as a standing assumption.

**Assumption 3.3.** For the source domain, if two points $s_i, s_j$ belong to different orbits, then $\mathrm{Pattern}(s_i) \neq \mathrm{Pattern}(s_j)$. The same condition holds for the target domain.

**Remark.** In $\mathbb{R}^d$, Assumption 3.3 holds generically: for point sets in general position, it is satisfied almost everywhere in the parameter space. A formal proof, given in Appendix A.4, shows that configurations violating the assumption form a measure-zero subset. It is important to note, however, that the assumption is not universal. There exist highly structured, non-generic point configurations for which it fails. We construct explicit pathological counterexamples in Appendix A.5. Together, these results delineate both the broad applicability of our framework and its precise theoretical boundary.

Under Assumption 3.3, two points belong to the same orbit if and only if they share the same distance pattern. This provides a practical scheme to extract the orbit decomposition solely from the cost matrix $C$, eliminating the need for any a prior specification of the group structure.

Once the orbit structure is identified, we can permute the points into a systematic pattern that greatly simplifies the subsequent analysis. We call this arrangement a reducible ordering. For the source domain, it is obtained by permuting the indices so that points belonging to the same orbit are indexed congruently modulo $m$:

$$\forall k \in \{1, \ldots, m\},$$
$$\forall s_i, s_j \in \{s_{m(a-1)+k} \mid a \in \{1, \ldots, |G|\}\}, \quad (7)$$
$$\exists g \in G \quad \text{s.t.} \quad s_j = g s_i.$$

The target domain is permuted analogously (modulo $n$).

Under this ordering, the permutation matrices $T_g^s$ and $T_g^t$ now act by cyclically permuting rows (resp. columns) within index classes modulo $m$ (resp. $n$). This action imposes a specific algebraic structure on any matrix that is invariant under the permutation matrices. The following proposition precisely characterizes this structure; its proof is given in Appendix A.6.

**Proposition 3.4.** *Suppose the points are arranged in a reducible ordering, and let $T_g^s, T_g^t$ be the associated permutation matrices. If a matrix $A \in \mathbb{R}^{m|G| \times n|G|}$ satisfies*

$$T_g^s A (T_g^t)^\top = A, \quad \forall g \in G,$$

*then, when partitioned into $|G| \times |G|$ blocks of size $m \times n$ as*

$$A = \begin{pmatrix} A^{1,1} & \cdots & A^{1,|G|} \\ \vdots & \ddots & \vdots \\ A^{|G|,1} & \cdots & A^{|G|,|G|} \end{pmatrix}, \quad A^{a,b} \in \mathbb{R}^{m \times n},$$

*the following properties hold:*

(a) *For each $k \in \{1, \ldots, m\}$ and any $a \in \{2, \ldots, |G|\}$, the $(m(a-1)+k)$-th row of $A$ is a permutation of its $k$-th row.*

(b) *For any $k, l$, define the $|G| \times |G|$ matrix $\widetilde{A}^{k,l}$ by $(\widetilde{A}^{k,l})_{a,b} = A_{k,l}^{a,b}$. Then every row sum of $\widetilde{A}^{k,l}$ equals every column sum.*

Furthermore, the probability measures become highly regular. They take the form:

$$p = (\alpha, \alpha, \ldots, \alpha)^\top, \quad q = (\beta, \beta, \ldots, \beta)^\top, \quad (8)$$

where $\alpha \in \mathbb{R}^m$ and $\beta \in \mathbb{R}^n$ are vectors repeated $|G|$ times.

Combining the structural description of Proposition 3.4 and the regularized measures (8), the permuted SG-ROT problem reduces to a substantially smaller problem on the orbit space. We obtain the following Orbit-Reduced SG-ROT (OR-SG-ROT) problem:

$$\min_{P^{1,1}, \ldots, P^{1,|G|} \in \mathbb{R}_+^{m \times n}} \sum_{b=1}^{|G|} \left( \langle C^{1,b}, P^{1,b} \rangle + \sum_{i=1}^m \sum_{j=1}^n \phi(P_{i,j}^{1,b}) \right)$$

$$\text{s.t.} \quad \sum_{b=1}^{|G|} P^{1,b} \mathbf{1}_n = \alpha, \quad \sum_{b=1}^{|G|} P^{1,b\top} \mathbf{1}_m = \beta.$$

$$(9)$$

A detailed proof of this reduction is provided in Appendix A.7.

## 4. Fast Algorithms for SG-ROT

In this section, we present efficient algorithms for solving SG-ROT, focusing on its two classical cases: the linear case (SG-LOT) and the entropic case (SG-EROT). Despite different regularizers, both problems share a common solution pipeline that exploits the underlying symmetry. This pipeline consists of three stages: (i) orbit identification and problem permutation; (ii) solving a orbit-reduced problem; and (iii) lifting the reduced solution back to the original scale. Figure 3 shows the overall workflow.

We first describe the stages common to both SG-LOT and SG-EROT. Orbit identification uses Assumption 3.3 to cluster points into orbits from the cost matrix alone. The points are then permuted into a reducible ordering, which brings the cost matrix and measures into the structured forms. After this permutation, the original SG-ROT problem(6) reduces to the OR-SG-ROT problem (9). Once the reduced problem is solved, the full invariant transport plan of the permuted problem is reconstructed using the structural consistency between the cost matrix and the final transport plan characterized in Proposition 3.4(a); a final inverse permutation yields the solution of the original problem.

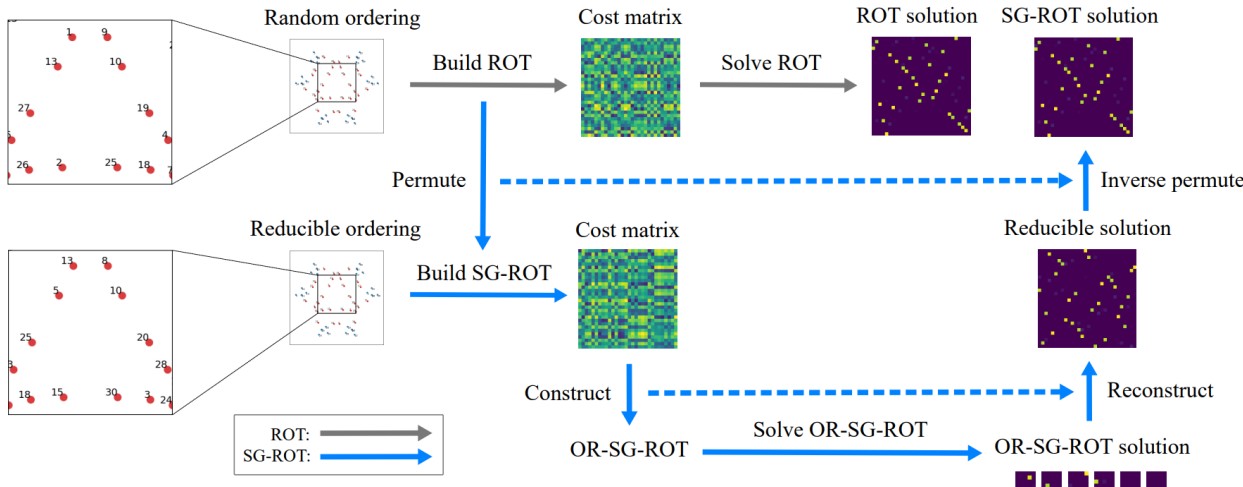

*Figure 3.* The workflow leverages group symmetry to transform the original high-dimensional problem into a lower-dimensional one. First, the cost matrix and measures are permuted into a reducible form, exposing the orbit structure. The problem is then reduced to OR-SG-ROT and solved on the orbit space. Finally, the solution is lifted and inverse-permuted to recover the full transport plan. This structured approach yields significant computational gains over direct methods.

## 4.1. Orbit Identification and Permutation

Under Assumption 3.3, points belonging to the same orbit produce identical rows (or columns) in the cost matrix $C$ when viewed as multisets. Consequently, the orbit structure can be recovered directly by comparing the rows and columns of $C$, and $C$ can be permuted accordingly into a reducible ordering. Notably, the number of orbits $|G|$ emerges automatically from these comparisons, requiring no prior knowledge of the group. In practice, we sort and group the rows and columns separately: rows (resp. columns) with identical distance patterns are assigned to the same orbit, and the corresponding row and column permutation matrices are constructed. These permutations are applied simultaneously to the cost matrix and to the source and target measures, preserving the equivalence of the transport problem. The entire procedure involves only sorting, grouping, and a single matrix permutation, with an overall time complexity of $O\big(mn|G|^2(\log(m|G|) + \log(n|G|))\big)$ for a cost matrix of size $m|G| \times n|G|$.

## 4.2. Fast Algorithms for OR-SG-ROT

Having reduced the original SG-ROT problem (6) to the OR-SG-ROT problem (9), we now address its numerical solution. The reduced problem shares the same mathematical structure in prior work (e.g., (Takeda et al., 2024)), whose algorithms can therefore be applied directly. We briefly outline the solution methods for its two main instances: the linear case (OR-SG-LOT) and the entropic case (OR-SG-EROT).

**OR-SG-LOT** For the linear case $\phi = 0$, the OR-SG-ROT problem reduces to OR-SG-LOT. It can be transformed into a standard OT problem of size $m \times n$, whose solution yields the optimal OR-SG-LOT plan.

**Proposition 4.1.** *Define the aggregated variable $S := \sum_{b=1}^{|G|} P^{1,b}$ and the reduced cost matrix $K \in \mathbb{R}^{m \times n}$ by*

$$K_{i,j} = \min_{1 \le b \le |G|} C_{i,j}^{1,b}.$$

*Then OR-SG-LOT is equivalent to the following linear optimal transport problem:*

$$\min_{S \in \mathbb{R}_+^{m \times n}} \quad \langle K, S \rangle, \tag{10}$$
$$s.t. \quad S\mathbf{1}_n = \alpha, \quad S^\top \mathbf{1}_m = \beta.$$

*Let $S^*$ be an optimal solution of (10). An optimal solution $\big\{P^{1,b*}\big\}_{b=1}^{|G|}$ of OR-SG-LOT is obtained by setting, for each $(i,j)$,*

$$P_{i,j}^{1,b*} = \begin{cases} S_{i,j}^*, & b = \min \operatorname{argmin}_{1 \le b' \le |G|} C_{i,j}^{1,b'}, \\ 0, & otherwise. \end{cases}$$

The equivalent problem (10) is a standard linear OT problem of size $m \times n$. Using an efficient network-simplex implementation (Ahuja et al., 1994), Tarjan (1997) showed that the worst-case time complexity is $O\big(mn(m + n) \log(m + n) \log((m + n)\|K\|_\infty)\big)$, where $\|K\|_\infty$ denotes the infinity norm of the matrix.

**OR-SG-EROT** For the entropic case $\phi(x) = \lambda x(\log x - 1)$, the OR-SG-ROT problem reduces to OR-SG-EROT. Its

solution can be obtained by solving a Fenchel dual problem (Rockafellar, 2015), which leads to a Sinkhorn-type iteration scheme.

**Proposition 4.2.** *The OR-SG-EROT problem admits the following Fenchel dual:*

$$\underset{u\in\mathbb{R}^m v\in\mathbb{R}^n}{\operatorname{argmax}} \langle u, \alpha\rangle + \langle v, \beta\rangle - \lambda \sum_{i=1}^{m}\sum_{j=1}^{n} p_i\, L_{i,j}\, q_j, \quad (11)$$

*where $p_i = \exp(u_i/\lambda), q_j = \exp(v_j/\lambda)$, and*

$$L_{i,j} := \sum_{b=1}^{|G|} \exp\Big(-\frac{C_{i,j}^{1,b}}{\lambda}\Big).$$

*Let $u^*, v^*$ be the optimal solution of (11). The primal optimal transport blocks are recovered from the dual variables*

$$P_{i,j}^{1,b^*} = p_i^* q_j^* \exp\Big(-\frac{C_{i,j}^{1,b}}{\lambda}\Big), \quad b = 1, \ldots, |G|.$$

The dual problem (11) can be solved efficiently by the Sinkhorn-Knopp algorithm (Takeda et al., 2024), which alternately updates $p_i \leftarrow \alpha_i/\sum_j L_{i,j}q_j$ and $q_j \leftarrow \beta_j/\sum_i L_{i,j}p_i$. Each Sinkhorn iteration then performs two matrix-vector products with $L$, costing $O(mn)$.

### 4.3. Solution Reconstruction

Once an optimal solution $\{P^{1,b}\}_{b=1}^{|G|}$ to the OR-SG-ROT problem is obtained, it is lifted back to a full transport plan $P \in \mathbb{R}^{m|G|\times n|G|}$ for the permuted SG-ROT problem. The reconstruction proceeds in two phases. First, the initial $m$ rows of $P$ are formed by placing the $|G|$ reduced matrices $P^{1,1}, \ldots, P^{1,|G|}$ side by side in the prescribed column order. Then, for each remaining row index $i > m$, let $k = i \mod m$; by Proposition 3.4(a) the $i$-th row of $P$ is a permutation of the $k$-th row, and the specific permutation is uniquely determined by the structure of the cost matrix $C$. This two-step procedure guarantees that the lifted matrix satisfies all invariance constraints of the original problem. Its time complexity is $O(mn|G|^2 \log(n|G|))$, dominated by the row-permutation lookups.

### 4.4. Total Computational Complexity Analysis

We now analyze the computational complexity of SG-LOT and SG-EROT. Following the pipeline, the cost of SG-LOT is dominated by two steps: solving the reduced problem (10) and the orbit-identification permutation. Overall, our method requires $O(mn(m + n) \log(m + n) \log((m+n)\|K\|_\infty) + mn|G|^2(\log(m|G|) + \log(n|G|)))$ time. In contrast, directly solving SG-LOT at the original $m|G| \times n|G|$ scale costs $O(mn(m + n)|G|^3 \log(m|G| + n|G|) \log((m|G| + n|G|)\|C\|_\infty))$, which is substantially

higher for large $|G|$. For SG-EROT, our pipeline reduces the cost to $O(Tmn + mn|G|^2(\log(m|G|) + \log(n|G|)))$, where $T$ is the number of Sinkhorn iterations. Solving the original $m|G| \times n|G|$ problem directly would require $O(T\,mn|G|^2)$ operations. The orbit-reduction therefore provides an speedup factor of roughly $|G|^2$ in the dominant iterative term.

The algorithm described above handles the general case where the points are given in a random order. In many practical scenarios, however, the source and target points may already follow a structured arrangement—such as the block ordering introduced in Appendix B—that naturally reflects the underlying symmetry. When such a favorable ordering is present, the orbit-identification step becomes trivial, and the reconstruction of the full transport plan can be carried out more efficiently. We detail these simplifications in Appendix B.

## 5. Experiments

We evaluate the proposed SG-ROT framework on synthetic and real datasets, with a focus on three aspects: computational efficiency, numerical correctness, and robustness under approximate or perturbed symmetries. In all experiments, the input is given as a cost matrix together with source and target measures; the goal is to recover the full transport plan and the corresponding objective value while recording the total runtime of the entire pipeline.

We compare the runtime of three methods: ROT (the baseline regularized OT), C-ROT (the cyclic-symmetry-aware solver of Takeda et al. (2024)), and our SG-ROT. For block ordered data we compare all three; for randomly ordered data we compare only ROT and SG-ROT (C-ROT is not applicable). Matrix accuracy is assessed via the $\ell_2$ norm of the difference between transport matrices. All reported results are averaged over multiple runs, with means and standard deviations shown. For the entropic case (EROT) we fix the regularization parameter $\lambda = 0.1$. Under random ordering, the orbit-identification and solution-lifting steps are accelerated on a GPU, while the remaining components run on CPU. All code is implemented in Python.

### 5.1. Synthetic Data with Exact Symmetry

We generate synthetic point sets in Euclidean space that strictly satisfy Assumption 3.1, covering five classical symmetry groups. Two-dimensional examples include the cyclic group $Z_3$ (order 3) and the dihedral group $D_3$ (order 6); three-dimensional examples include the tetrahedral $T$ (order 12), octahedral $O$ (order 24), and icosahedral $I$ (order 60) groups. Each dataset is constructed by first generating a Gaussian point cluster, then applying all group transformations to it, which naturally yields a block ordering. For

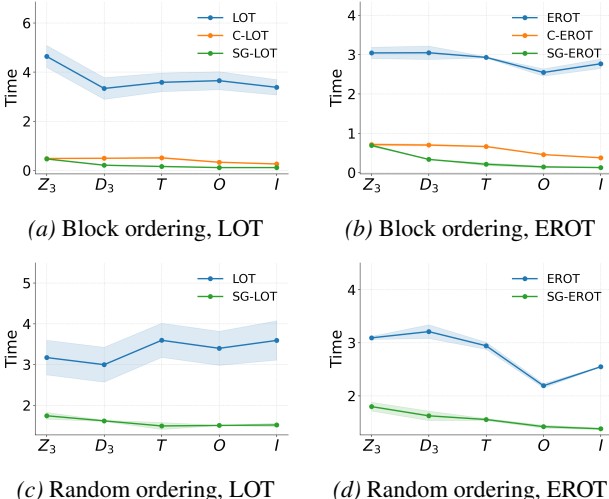

*(a)* Block ordering, LOT   *(b)* Block ordering, EROT

*(c)* Random ordering, LOT   *(d)* Random ordering, EROT

*Figure 4.* Runtime comparison of different methods under varying symmetry groups for 4800 points. Shaded regions indicate standard deviation across runs.

instance, for $Z_3$ we start with a cluster of 1,600 points, apply two successive $120°$ rotations, and obtain a final set of 4,800 points. The corresponding probability measure is built by sampling a single random positive vector for the initial cluster, repeating it cyclically according to the group order, and normalizing the whole vector to sum to one. The other groups follow the same procedure, with initial cluster sizes of 800 ($D_3$), 400 ($T$), 200 ($O$), and 80 ($I$). For experiments under random ordering, we simply shuffle the point indices before applying the algorithms. To further assess scalability, we also create larger versions of these datasets (e.g., 9,600 points) using the same generation protocol.

Figure 4 summarizes the runtime results. Under block ordering, SG-ROT exploits the underlying symmetry more thoroughly than C-ROT, especially for larger and more complex groups, leading to a clear reduction in computational time. Even under random ordering, SG-ROT still runs faster than the direct ROT baseline, with the speedup growing as the group order increases. Table 1 further verifies numerical correctness for the octahedral group $O$: the objective values and transport plans produced by SG-ROT match those of direct ROT to within machine precision. All synthetic datasets in our study satisfy Assumption 3.3, and the consistent success of orbit identification across groups also provides empirical support that this assumption is naturally met in generic symmetric configurations. Additional analyses on scaling behavior and resource usage are provided in Appendix C.1. Complete numerical results across different symmetry groups are provided in Appendix C.2.

## 5.2. Image Data with Approximate Symmetry

To assess performance on real-world data, we apply our method to the NYU Symmetric Image Dataset (Cicconet et al., 2017), which contains images that approximately exhibit dihedral symmetry $D_2$. The dataset includes six such images. Figure 5 shows representative examples. We consider all pairwise combinations of these images, giving a total of 15 image-pair experiments. Each image is resized to two resolutions, $60 \times 80$ and $80 \times 120$; pixel intensities are normalized to form probability measures. We compare EROT, C-EROT, and SG-EROT under the block-ordered setting, and EROT versus SG-EROT under random ordering.

The results are summarized in Table 2. Under block ordering, SG-EROT successfully leverages the underlying approximate symmetry, obtaining a lower runtime than direct EROT while maintaining nearly identical objective values. Under random ordering, SG-EROT still runs faster than the EROT baseline, though small deviations in both objective values and transport-matrix distances appear—a consequence of the approximate rather than exact symmetry in the real image data.

## 5.3. Image Data with Perturbed Symmetry

To assess the robustness of SG-EROT when the underlying symmetry is only approximate, we again employ the 15 image pairs from the NYU dataset that exhibit approximate dihedral symmetry $D_2$. We progressively break this symmetry by adding Gaussian noise to the upper-left quadrant of each image, using noise variances from the set $\{0, 20, 40, 80, 160, 320, 640\}$. After perturbation, images are resized to $60 \times 80$ and their pixel intensities normalized to unit total measure, forming the probability measures. We then compare EROT and SG-EROT across increasing noise levels, measuring the differences in objective values and transport matrix distances.

Figure 6 illustrates the sensitivity of SG-EROT to symmetry violations. As the noise level increases, the discrepancy between the SG-EROT and direct EROT solutions grows gradually. Importantly, this growth is smooth and shows no abrupt jumps under mild perturbations, indicating that the method remains stable and degrades gracefully as the underlying symmetry is progressively broken.

## 5.4. Molecular Graph Data with Exact Symmetry

To further evaluate our method on real-world structured data, we conduct experiments on molecular graphs. Following the pipeline of (Togninalli et al., 2019), we represent molecules as graphs and construct cost matrices using Weisfeiler–Lehman node features with Hamming distance. For highly symmetric molecules, when atoms are ordered according to the block ordering, the resulting cost matrix

*Table 1.* Numerical results on synthetic data with octahedral group $O$ for 4,800 and 9,600 points.

| Ordering | Method | 4,800 points | | | 9,600 points | | |
|---|---|---|---|---|---|---|---|
| | | Obj.value | Solution distance | Time (sec.) | Obj.value | Solution distance | Time (sec.) |
| Block | LOT | $0.2268 \pm 0.0000$ | – | $3.6485 \pm 0.3661$ | $0.1967 \pm 0.0000$ | – | $16.8215 \pm 1.5525$ |
| | C-LOT | $0.2268 \pm 0.0000$ | $0.0000 \pm 0.0000$ | $0.3289 \pm 0.0155$ | $0.1967 \pm 0.0000$ | $0.0000 \pm 0.0000$ | $1.5622 \pm 0.0142$ |
| | SG-LOT | $0.2268 \pm 0.0000$ | $0.0000 \pm 0.0000$ | $0.1112 \pm 0.0059$ | $0.1967 \pm 0.0000$ | $0.0000 \pm 0.0000$ | $0.4696 \pm 0.0026$ |
| | EROT | $-0.9248 \pm 0.0000$ | – | $2.5442 \pm 0.0916$ | $-1.0696 \pm 0.0000$ | – | $11.6585 \pm 0.2908$ |
| | C-EROT | $-0.9248 \pm 0.0000$ | $0.0000 \pm 0.0000$ | $0.4577 \pm 0.0041$ | $-1.0696 \pm 0.0000$ | $0.0000 \pm 0.0000$ | $2.0111 \pm 0.0107$ |
| | SG-EROT | $-0.9248 \pm 0.0000$ | $0.0000 \pm 0.0000$ | $0.1457 \pm 0.0200$ | $-1.0696 \pm 0.0000$ | $0.0000 \pm 0.0000$ | $0.6789 \pm 0.0029$ |
| Random | LOT | $0.2153 \pm 0.0000$ | – | $3.3981 \pm 0.4171$ | $0.1794 \pm 0.0000$ | – | $18.1031 \pm 1.9482$ |
| | SG-LOT | $0.2153 \pm 0.0000$ | $0.0000 \pm 0.0000$ | $1.5056 \pm 0.0036$ | $0.1794 \pm 0.0000$ | $0.0000 \pm 0.0000$ | $4.1651 \pm 0.3262$ |
| | EROT | $-0.9377 \pm 0.0000$ | – | $2.1895 \pm 0.0398$ | $-1.0771 \pm 0.0000$ | – | $10.4986 \pm 0.1323$ |
| | SG-EROT | $-0.9377 \pm 0.0000$ | $0.0000 \pm 0.0000$ | $1.4189 \pm 0.0274$ | $-1.0771 \pm 0.0000$ | $0.0000 \pm 0.0000$ | $4.1449 \pm 0.3813$ |

*Table 2.* Numerical results on real images with approximate dihedral group $D_2$ for 4,800 and 9,600 points.

| Ordering | Method | 4,800 points | | | 9,600 points | | |
|---|---|---|---|---|---|---|---|
| | | Obj.value | Solution distance | Time (sec.) | Obj.value | Solution distance | Time (sec.) |
| Block | EROT | $-0.9789 \pm 0.0784$ | – | $12.9147 \pm 3.2836$ | $-1.1366 \pm 0.0530$ | – | $50.5090 \pm 10.5048$ |
| | C-EROT | $-0.9876 \pm 0.0823$ | $0.0010 \pm 0.0005$ | $3.9716 \pm 0.1673$ | $-1.1761 \pm 0.0443$ | $0.0005 \pm 0.0003$ | $15.9426 \pm 0.2879$ |
| | SG-EROT | $-0.9909 \pm 0.0971$ | $0.0011 \pm 0.0005$ | $0.8639 \pm 0.0967$ | $-1.2045 \pm 0.0619$ | $0.0006 \pm 0.0003$ | $4.1361 \pm 0.3335$ |
| Random | EROT | $-0.9789 \pm 0.0784$ | – | $13.9812 \pm 3.4573$ | $-1.1366 \pm 0.0530$ | – | $50.3322 \pm 9.1211$ |
| | SG-EROT | $-0.9909 \pm 0.0971$ | $0.0011 \pm 0.0005$ | $2.1840 \pm 0.1705$ | $-1.2045 \pm 0.0619$ | $0.0006 \pm 0.0003$ | $7.5900 \pm 0.1929$ |

*Table 3.* Numerical results on graph matching.

| Method | Obj.value | Solution distance | Time (ms) |
|---|---|---|---|
| EROT | -3.045983 ± 0.000161 | – | $0.5850 \pm 0.1590$ |
| SG-EROT | -3.045982 ± 0.000161 | $0.0000 \pm 0.0000$ | $0.2909 \pm 0.0412$ |

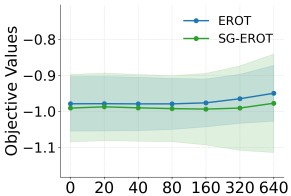

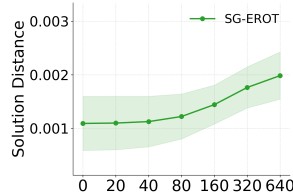

*(a)* Objective values vs noise     *(b)* Solution distance vs noise

*Figure 6.* Robustness to symmetry perturbations. Shaded regions indicate standard deviation across runs.

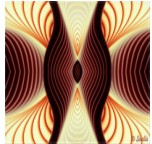 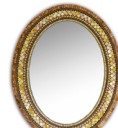 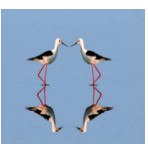 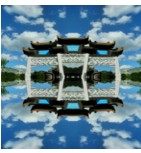

*Figure 5.* Example images from the NYU Symmetric Image Dataset. The pixel grids approximately exhibit geometric symmetry under dihedral group $D_2$.

directly exhibits a block-circulant structure whose block size equals the group order. We select eight symmetric compounds: hexamethylbenzene, hexahydroxybenzene, hexamethoxybenzene, hexaaminobenzene, hexaformylbenzene, hexaacetylbenzene, hexacarboxybenzene, and hexabenzylbenzene. We evaluate all 28 pairwise comparisons using both EROT and SG-EROT.

The results are summarized in Table 3. SG-EROT preserves the same objective value as standard EROT up to numerical precision. This indicates that the proposed method generalizes well to real molecular graph data.

## 6. Discussions

This work introduces SG-ROT, a general framework that exploits intrinsic symmetries in data to reduce the computational cost of optimal transport. We conclude by outlining several directions for future research. (1) Our algorithm relies on Assumption 3.1. When measures only approximately satisfy this assumption, the orbit-reduced formulation yields an approximate transport solution. An open theoretical question is to characterize the stability of this approximation with respect to symmetry perturbations and to quantify how the transport error depends on the degree of asymmetry. (2) While this work establishes a general theoretical connection between finite group actions and optimal transport, its practical utility would benefit from more extensive experimental evaluations across diverse real-world applications. Testing the framework in domains such as molecular structure align-

ment, symmetric object matching, or periodic time-series analysis could reveal both its strengths and potential limitations in practice. (3) Another promising direction is to extend the proposed framework to Gromov–Wasserstein optimal transport, which could enable symmetry-aware acceleration schemes for this important variant. Since Gromov–Wasserstein distances are widely used for comparing metric measure spaces, incorporating symmetry reductions could significantly broaden the computational feasibility of these methods. (4) Finally, exploring how SG-ROT can be integrated with existing optimal-transport acceleration methods that leverage specific structural properties of the data—such as low-rank, sparse, or hierarchical structures—may lead to further computational gains. We leave the exploration of these directions to future work.

## 7. Conclusion

This paper presents SG-ROT, a framework that accelerates optimal transport by fully exploiting intrinsic data symmetries. We show that the original high-dimensional problem can be rigorously reduced to a smaller-scale optimal transport problem without loss of optimality. Our method is robust to arbitrary input orderings and does not require any special initial arrangement of the data. Experiments confirm that SG-ROT effectively leverages symmetry and maintains invariance under arbitrary permutations of the input. To our knowledge, this is the first work that systematically employs symmetry groups to accelerate optimal transport, opening a direction for future research in symmetry-aware computational optimal transport.

## Acknowledgements

This work was supported by the National Key Research and Development Program of China (2021YFA1003000).

## Impact Statement

This work aims to advance optimal transport by developing a method that uses symmetry in data to speed up computation. As an algorithmic improvement, it makes optimal transport faster for various applications, such as point cloud registration and symmetric shape matching. This work has no ethical concerns as far as we know.

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

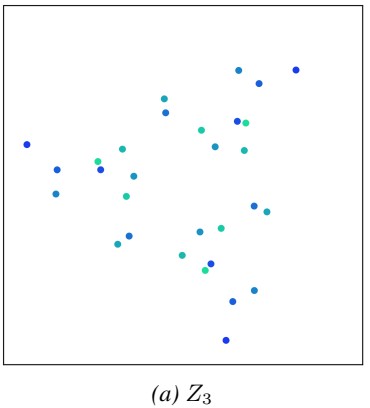

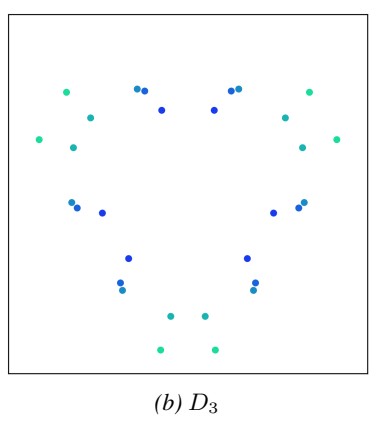

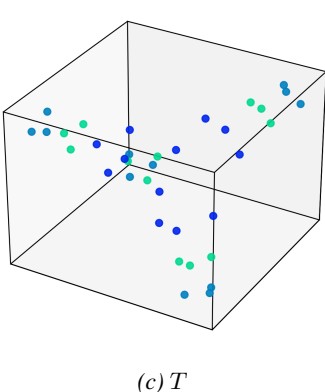

*(a) $Z_3$*        *(b) $D_3$*        *(c) $T$*

*Figure 7.* Examples of orbit decompositions under free group actions. (a) A point set with three-fold rotational symmetry, admitting a free action of the cyclic group $Z_3$ (no point is fixed by a non-identity rotation). (b) A point set symmetric under the full dihedral group $D_3$ of a triangle; the action is free, with no fixed points under any reflection or non-trivial rotation. (c) A three-dimensional set possessing the 12 rotational symmetries of a regular tetrahedron (the tetrahedral group $T$); the action is again free. In all cases, points belonging to the same orbit share the same color.

## A. Proofs and Examples

### A.1. Examples of Symmetry Groups and Orbit Decomposition

This appendix provides visual examples of point sets under free group actions, complementing the discussion in Section 2.2. We consider three cases: (i) two-dimensional sets with free actions of the cyclic group $Z_3$ and the dihedral group $D_3$; and (ii) a three-dimensional set with a free action of the tetrahedral group $T$. Points belonging to the same orbit are rendered with the same color intensity (see Figure 7). This visualization clearly illustrates the orbit decomposition resulting from the free group actions.

### A.2. Proof of Lemma 3.2

*Proof.* Let $P^*$ be an optimal solution of SG-ROT. Applying the group averaging operator to $P^*$, we define

$$\overline{P} = \frac{1}{|G|} \sum_{g \in G} T_g^s P^* (T_g^t)^\top.$$

We show that $\overline{P}$ is also an optimal solution of SG-ROT. We first verify feasibility. We have

$$\overline{P} \mathbf{1}_{n|G|} = \frac{1}{|G|} \sum_{g \in G} T_g^s P^* (T_g^t)^\top \mathbf{1}_{n|G|}$$

$$= \frac{1}{|G|} \sum_{g \in G} T_g^s P^* \mathbf{1}_{n|G|}$$

$$= \frac{1}{|G|} \sum_{g \in G} T_g^s p.$$

Since all points within the same orbit share identical measures and the group action permutes points only within each orbit, we have $T_g^s p = p$ for all $g \in G$. Hence,

$$\overline{P} \mathbf{1}_{n|G|} = \frac{1}{|G|} \sum_{g \in G} p = p.$$

By the same argument, $\overline{P}^\top \mathbf{1}_{m|G|} = q$, and therefore $\overline{P}$ is feasible for SG-ROT.

Next, we consider the objective function. By the invariance of the cost matrix $C$ (4), we have

$$
\begin{aligned}
\langle C, \overline{P} \rangle &= \left\langle C, \frac{1}{|G|} \sum_{g \in G} T_g^s P^* (T_g^t)^\top \right\rangle \\
&= \frac{1}{|G|} \sum_{g \in G} \langle C, T_g^s P^* (T_g^t)^\top \rangle \\
&= \frac{1}{|G|} \sum_{g \in G} \langle T_{g^{-1}}^s C (T_{g^{-1}}^t)^\top, P^* \rangle \\
&= \frac{1}{|G|} \sum_{g \in G} \langle C, P^* \rangle \\
&= \langle C, P^* \rangle.
\end{aligned}
$$

Moreover, since $\phi$ is a convex function, Jensen's inequality implies

$$
\begin{aligned}
\sum_{i=1}^{|S|} \sum_{j=1}^{|T|} \phi(\overline{P}_{i,j}) &= \sum_{i=1}^{|S|} \sum_{j=1}^{|T|} \phi\left( \left( \left( \frac{1}{|G|} \sum_{g \in G} T_g^s P^* (T_g^t)^\top \right) \right)_{i,j} \right) \\
&\leq \frac{1}{|G|} \sum_{g \in G} \sum_{i=1}^{|S|} \sum_{j=1}^{|T|} \phi\left( (T_g^s P^* (T_g^t)^\top)_{i,j} \right) \\
&= \frac{1}{|G|} \sum_{g \in G} \sum_{i=1}^{|S|} \sum_{j=1}^{|T|} \phi(P_{i,j}^*) \\
&= \sum_{i=1}^{|S|} \sum_{j=1}^{|T|} \phi(P_{i,j}^*).
\end{aligned}
$$

Therefore, $\overline{P}$ is also an optimal solution of SG-ROT.

Finally, for any $g' \in G$, since $G = g'G$, we have

$$
\begin{aligned}
T_{g'}^s \overline{P} T_{g'}^{t\,\top} &= T_{g'}^s \frac{1}{|G|} \sum_{g \in G} T_g^s P^* (T_g^t)^\top (T_{g'}^t)^\top \\
&= \frac{1}{|G|} \sum_{g \in G} T_{g'}^s T_g^s P^* (T_g^t)^\top (T_{g'}^t)^\top \\
&= \frac{1}{|G|} \sum_{g \in G} T_{gg'}^s P^* (T_{gg'}^t)^\top \\
&= \frac{1}{|G|} \sum_{g \in G} T_g^s P^* (T_g^t)^\top = \overline{P}.
\end{aligned}
$$

Thus, $\overline{P}$ is invariant under the group action, which completes the proof. $\square$

### A.3. Proof of Equivalence Between SG-ROT and Its Symmetric Form

*Proof.* The constrained problem (6) has a smaller feasible set than the original SG-ROT problem, so its optimal value is at least as large. By Lemma 3.2, the original original SG-ROT problem itself has a group-invariant optimal plan, which is feasible for the constrained problem and yields the same objective value. Therefore the two problems have identical optimal values, and any optimal solution of (6) is also optimal for the original SG-ROT problem. $\square$

## A.4. Proof of Generic Validity of Assumption 3.3

*Proof.* We show that the set of point configurations violating Assumption 3.3—i.e., where distinct orbits produce identical distance patterns—has Lebesgue measure zero in the space of all symmetric configurations. The argument proceeds by parameterizing symmetric configurations, constructing a polynomial that vanishes exactly when the assumption fails, and then appealing to the measure-zero property of nonzero polynomial zero sets.

First, we parameterize the symmetric point sets. Let $x_1, \ldots, x_m \in \mathbb{R}^d$ and $y_1, \ldots, y_n \in \mathbb{R}^d$ be representatives of the source and target orbits, respectively. Applying the full group $G$ to these representatives generates the whole point sets $S$ and $T$. Hence any symmetric configuration is uniquely described by the tuple

$$(x_1, \ldots, x_m, y_1, \ldots, y_n) \in \mathbb{R}^{(m+n)d},$$

which we treat as the parameter space.

Next, we encode the condition of identical distance patterns as polynomial equations. For a source orbit representative $x_i$, consider the multiset of squared distances to all target points:

$$A_i := \{ \, \|x_i - gy_j\|^2 \mid g \in G, \ j = 1, \ldots, n \, \}.$$

Each element of $A_i$ is a polynomial in the coordinates of the representatives. Two source representatives $x_i, x_{i'}$ have identical distance patterns if and only if $A_i = A_{i'}$. To test equality of multisets, we use elementary symmetric polynomials: let $e_k(A_i)$ be the $k$-th elementary symmetric polynomial of the numbers in $A_i$, and set

$$f_{i,i'} := \sum_{k=1}^{n|G|} \big( e_k(A_i) - e_k(A_{i'}) \big)^2.$$

Then $f_{i,i'} = 0$ precisely when $A_i = A_{i'}$.

Analogously, for target representatives define

$$B_j := \{ \, \|gx_i - y_j\|^2 \mid g \in G, \ i = 1, \ldots, m \, \},$$

and

$$g_{j,j'} := \sum_{k=1}^{m|G|} \big( e_k(B_j) - e_k(B_{j'}) \big)^2,$$

so that $g_{j,j'} = 0$ if and only if $B_j = B_{j'}$.

Now collect all these constraints into a single polynomial

$$P := \Big( \prod_{1 \leq i < i' \leq m} f_{i,i'} \Big) \Big( \prod_{1 \leq j < j' \leq n} g_{j,j'} \Big).$$

By construction, $P(x_1, \ldots, y_n) = 0$ if and only if there exist distinct source orbits or distinct target orbits with identical distance patterns—that is, exactly when Assumption 3.3 fails.

Finally, because $P$ is a nonzero polynomial (generic configurations satisfy the assumption, as illustrated by the examples in the main text), its zero set has Lebesgue measure zero in $\mathbb{R}^{(m+n)d}$ (Mityagin, 2015). Consequently, the set of symmetric point configurations that violate Assumption 3.3 is a measure-zero subset of the parameter space; in other words, the assumption holds for almost every configuration. $\qquad\square$

## A.5. A Constructed Counterexample Illustrating the Necessity of Assumption 3.3

This appendix presents a carefully constructed counterexample to demonstrate the necessity of Assumption 3.3 introduced in Section 3.2. The example shows that when the assumption is violated, distinct orbits can become fundamentally indistinguishable based solely on the pairwise distances in the cost matrix, thereby establishing a theoretical boundary for orbit identification from distance data.

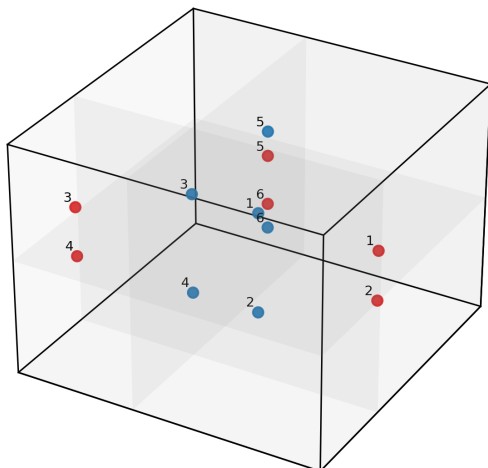

*Figure 8.* Geometric configuration of the counterexample point sets in $\mathbb{R}^3$. Red points represent the source set $S$; blue points represent the target set $T$.

The following counterexample is inspired by constructions in distance geometry (Boutin & Kemper, 2004; Mucherino et al., 2012). Consider $\mathbb{R}^3$. The source point set is

$$S = \{(2.5, 0.5, 0.5), \ (2.5, 0.5, -0.5), \ (-1.5, -0.5, 0.5), \ (-1.5, -0.5, -0.5), \ (0, 2, 0.5), \ (0, 2, -0.5)\},$$

and the target set is

$$T = \{(1, 0, 1), \ (1, 0, -1), \ (0, 0, 1), \ (0, 0, -1), \ (0, 2, 1), \ (0, 2, -1)\}.$$

Figure 8 illustrates the geometric configuration of these points.

Here, the symmetry group $G \cong \mathbb{Z}_2$ is generated by reflection across the plane $z = 0$. Its action on both $S$ and $T$ is free.

A direct computation shows

$$\text{Pattern}(s_1) = \text{Pattern}(s_2) = \text{Pattern}(s_3) = \text{Pattern}(s_4),$$

while the points $s_1, s_2, s_3, s_4$ do not lie in the same orbit. Hence, Assumption 3.3 fails for this construction.

### A.6. Proof of Proposition 3.4

*Proof.* (a) This follows directly from the free and transitive action of the group on each orbit.

(b) The group action is free, so the entries of $\widetilde{A}^{k,l}$ are permuted transitively and without repetition. Hence, each row and each column contains the same multiset of elements. It follows that all row sums equal all column sums. □

### A.7. Proof of Equivalence Between OR-SG-ROT and SG-ROT

*Proof.* We first consider the objective function. By Proposition 3.4(a), we have

$$
\langle C, P \rangle + \sum_{i=1}^{m|G|} \sum_{j=1}^{n|G|} \phi(P_{i,j})
$$

$$
= \sum_{a=1}^{|G|} \sum_{b=1}^{|G|} \langle C^{a,b}, P^{a,b} \rangle + \sum_{a=1}^{|G|} \sum_{b=1}^{|G|} \sum_{i=1}^{m} \sum_{j=1}^{n} \phi\left(P_{i,j}^{a,b}\right)
$$

$$
= |G| \sum_{b=1}^{|G|} \left( \langle C^{1,b}, P^{1,b} \rangle + \sum_{i=1}^{m} \sum_{j=1}^{n} \phi\left(P_{i,j}^{1,b}\right) \right).
$$

We next examine the measures constraints. By Proposition 3.4(a), we have

$$P\mathbf{1}_{n|G|} = \left(\sum_{b=1}^{|G|} P^{1,b}\mathbf{1}_n, \sum_{b=1}^{|G|} P^{2,b}\mathbf{1}_n, \ldots, \sum_{b=1}^{|G|} P^{|G|,b}\mathbf{1}_n\right)^{\top}$$

$$= \left(\sum_{b=1}^{|G|} P^{1,b}\mathbf{1}_n, \sum_{b=1}^{|G|} P^{1,b}\mathbf{1}_n, \ldots, \sum_{b=1}^{|G|} P^{1,b}\mathbf{1}_n\right)^{\top}.$$

On the other hand, by Proposition 3.4(b), we have

$$\sum_{i=1}^{m|G|} P_{i,1} = \sum_{k=1}^{m}\sum_{a=1}^{|G|} P_{m(a-1)+k,1} = \sum_{k=1}^{m}\sum_{b=1}^{|G|} P_{k,n(b-1)+1} = \sum_{b=1}^{|G|}\sum_{k=1}^{m} P_{k,1}^{1,b}.$$

Consequently,

$$P^{\top}\mathbf{1}_{m|G|} = \left(\sum_{b=1}^{|G|} P^{1,b^{\top}}\mathbf{1}_m, \sum_{b=1}^{|G|} P^{1,b^{\top}}\mathbf{1}_m, \ldots, \sum_{b=1}^{|G|} P^{1,b^{\top}}\mathbf{1}_m\right)^{\top}.$$

The constant factor $|G|$ in the objective does not affect the optimal solution and can be omitted. In summary, the permuted SG-ROT can thus be transformed into OR-SG-ROT (9). $\square$

## B. Three Kinds of Point Ordering

This appendix compares three point orderings: random ordering, reducible ordering, and block ordering. The main advantage of block ordering is that it simplifies orbit identification and solution reconstruction. Figure 9 shows the resulting cost matrix structures.

### B.1. Random Ordering

A random ordering refers to point sets that only satisfy Assumption 3.1, with no prescribed arrangement of points within orbits. Consequently, the induced cost matrix shows minimal visible structure and appears essentially unstructured (see Figure 9a).

### B.2. Reducible Ordering

A reducible ordering is obtained by taking point sets that satisfy Assumptions 3.1 and 3.3 and then permuting them into a systematic arrangement. For the source domain, this means that for each $k \in \{1, \ldots, m\}$, the points

$$\{s_{m(a-1)+k} \mid a \in \{1, \ldots, |G|\}\}$$

all belong to the same orbit; an analogous condition holds for the target domain. This ordering induces a repeating pattern in the cost matrix: rows with indices congruent modulo $m$ are identical up to a permutation, and similarly for columns modulo $n$. As a result, the matrix appears much more uniform and regularly patterned than in the random case (see Figure 9b).

### B.3. Block Ordering

The block ordering is a special case of the reducible ordering that imposes an additional algebraic consistency: the indexing of points within each orbit must correspond directly to a fixed enumeration of the group elements. Formally, for every $g \in G$, there exists an index $a \in \{1, \ldots, |G|\}$ such that

$$gs_k = s_{m(a-1)+k}.$$

The same enumeration is used for the target domain. This consistency forces the cost matrix to exhibit an explicit $|G| \times |G|$ block structure, where the arrangement of blocks reflects the group multiplication table (see Figure 9c).

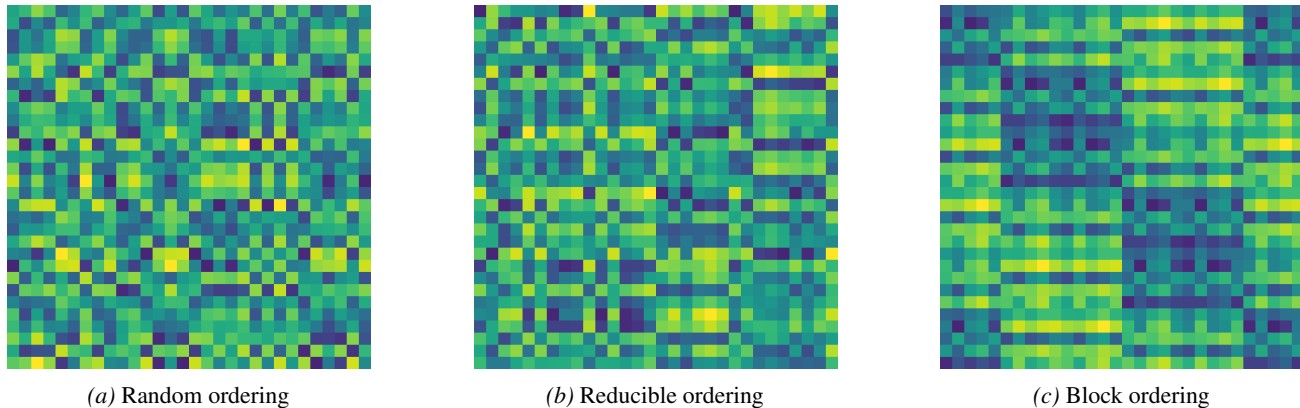

| (a) Random ordering | (b) Reducible ordering | (c) Block ordering |

*Figure 9.* Schematic illustrations of cost matrices induced by a $D_3$ symmetry between the source and target domains, with $m = n = 5$. Panel (a) shows the Random case, panel (b) the reducible ordering, and panel (c) the block ordering. Since $D_3$ has order 6, the block ordering exhibits a $6 \times 6$ block structure. Moreover, as $D_3$ contains a cyclic subgroup of order 3, the block structure is further visualized as a $3 \times 3$ cyclic sub-block.

When the points are given in a block ordering, the cost matrix exhibits an explicit $|G| \times |G|$ block structure, but the specific arrangement of these blocks—i.e., how the group elements correspond to row and column indices—is not known a priori. The following two steps exploit this block structure for acceleration.

**Recovering the block permutation.**   We first determine how blocks are permuted across rows. Taking the blocks in the first row as reference templates, we extract each block to create a unique signature and build a mapping from signatures to column indices. For every block in the remaining rows, we compute its signature and look up the corresponding reference position via this mapping. When multiple reference blocks share the same signature, we track already-used indices within the same row to ensure a consistent permutation per row. This procedure recovers the full block-permutation matrix, which encodes the action of the group on the matrix indices. Its time complexity is $O(|G|^2 mn + |G|^3)$: constructing reference signatures costs $O(|G|mn)$, scanning the $|G|^2$ block positions costs $O(|G|^2 mn)$, and in the worst case matching repeated signatures adds an extra $O(|G|^3)$ factor.

**Fast solution reconstruction using the known permutation.**   Once the block-permutation matrix is known, reconstructing the full transport plan reduces to a systematic rearrangement of the $|G|$ solved blocks $\{P^{1,b}\}_{b=1}^{|G|}$. We first stack these blocks into a three-dimensional array of shape $|G| \times m \times n$. Using the precomputed permutation (converted to zero-based indexing), we perform an advanced indexing operation that selects, for each row of blocks, the appropriate column block according to the permutation. This produces a four-dimensional array of shape $|G| \times |G| \times m \times n$, which is then permuted and reshaped into the final $(|G|m) \times (|G|n)$ matrix. The entire reconstruction step is a single pass over all $|G|^2$ blocks and costs $O(|G|^2 mn)$ time.

Thus, under block ordering the computational overhead consists of a one-time recovery of the block permutation, after which both problem reduction and solution reconstruction become straightforward indexing operations.

## C. More Experimental Results

### C.1. Scaling Analysis under D$_3$

This appendix evaluates the scalability of our method under the dihedral group $D_3$ by increasing the number of points from 4,800 to 14,400, considering both block and random orderings. Table 4 reports runtime, Table 5 reports memory usage, and Table 6 provides a detailed breakdown of SG-ROT methods under random ordering. In Table 6, the "Cost" column corresponds to computing the full cost matrix by scaling the OR-SG-ROT solution. Results show that SG-ROT consistently outperforms baselines in both runtime and memory, with the advantage growing with problem size. The decomposition reveals that permutation and reconstruction dominate the overhead, while the optimization step remains efficient.

*Table 4.* Runtime comparison under $D_3$ symmetry (sec.).

| Ordering | Method | 4800 | 7200 | 9600 | 12000 | 14400 |
|---|---|---|---|---|---|---|
| Block | LOT | 2.58 | 11.35 | 17.58 | 34.64 | 37.11 |
| | C-LOT | 0.39 | 1.02 | 1.91 | 3.12 | 4.70 |
| | SG-LOT | 0.16 | 0.38 | 0.71 | 1.12 | 1.79 |
| | EROT | 2.66 | 7.02 | 10.59 | 18.08 | 23.46 |
| | C-EROT | 0.53 | 1.40 | 2.63 | 4.19 | 6.62 |
| | SG-EROT | 0.27 | 0.59 | 1.08 | 1.82 | 2.90 |
| Random | LOT | 2.68 | 6.04 | 12.28 | 20.43 | 33.10 |
| | SG-LOT | 1.63 | 2.80 | 4.41 | 6.70 | 17.04 |
| | EROT | 2.64 | 5.75 | 9.24 | 16.21 | 24.07 |
| | SG-EROT | 1.53 | 2.59 | 4.12 | 6.43 | 16.88 |

*Table 5.* Memory comparison under $D_3$ symmetry. For random ordering, numbers are shown as CPU / GPU (MB).

| Ordering | Method | 4800 | 7200 | 9600 | 12000 | 14400 |
|---|---|---|---|---|---|---|
| Block | LOT | 811.6 | 1828.4 | 3594.1 | 5631.7 | 7641.0 |
| | C-LOT | 446.5 | 985.7 | 1822.3 | 2869.9 | 4179.1 |
| | SG-LOT | 389.7 | 912.0 | 1598.7 | 2500.8 | 3646.3 |
| | EROT | 718.4 | 1624.6 | 2894.9 | 4518.2 | 6513.4 |
| | C-EROT | 467.1 | 1061.1 | 1898.2 | 2970.5 | 4277.6 |
| | SG-EROT | 352.3 | 880.5 | 1622.2 | 2544.1 | 3643.0 |
| Random | LOT | 809.0 | 1827.1 | 3251.7 | 5612.6 | 8045.4 |
| | SG-LOT | 709.4/483.4 | 1109.5/1025.1 | 1857.2/1847.2 | 2727.4/2862.6 | 4756.4/4099.5 |
| | EROT | 719.1 | 1618.9 | 2891.0 | 4523.1 | 6517.4 |
| | SG-EROT | 307.9/483.4 | 735.9/1025.1 | 958.1/1847.2 | 1494.8/2862.6 | 2155.8/4099.5 |

*Table 6.* Runtime breakdown for SG-ROT under random ordering (sec.).

| Method | Points | Total Time | Permute | Solve OR-SG-ROT | Cost | Reconstruct and inverse permute |
|---|---|---|---|---|---|---|
| SG-LOT | 4800 | 1.63 | 0.78 | 0.09 | 0.02 | 0.74 |
| | 7200 | 2.80 | 1.17 | 0.20 | 0.05 | 1.37 |
| | 9600 | 4.41 | 1.71 | 0.36 | 0.09 | 2.25 |
| | 12000 | 6.70 | 2.31 | 0.63 | 0.18 | 3.59 |
| | 14400 | 17.04 | 10.99 | 0.96 | 0.26 | 4.83 |
| SG-EROT | 4800 | 1.53 | 0.66 | 0.10 | 0.08 | 0.69 |
| | 7200 | 2.59 | 0.90 | 0.18 | 0.17 | 1.33 |
| | 9600 | 4.12 | 1.31 | 0.35 | 0.32 | 2.14 |
| | 12000 | 6.43 | 1.87 | 0.60 | 0.50 | 3.45 |
| | 14400 | 16.88 | 10.47 | 0.87 | 0.83 | 4.70 |

## C.2. Results Across Different Symmetry Groups

This appendix presents additional numerical results for the four symmetry groups $Z_3, D_3, T, I$ discussed in Section 5.1, covering datasets of two sizes: 4,800 and 9,600 points. All results are summarized in Table 7. The experiments demonstrate that our method consistently reduces computational cost across different settings.

*Table 7.* Numerical results on synthetic datasets with different symmetry groups for 4,800 and 9,600 points.

| Group | Ordering | Method | 4,800 points | | | 9,600 points | | |
|---|---|---|---|---|---|---|---|---|
| | | | Obj.value | Solution distance | Time (sec.) | Obj.value | Solution distance | Time (sec.) |
| $Z_3$ | Block | LOT | $0.077 \pm 0.000$ | – | $4.635 \pm 0.445$ | $0.064 \pm 0.000$ | – | $18.070 \pm 0.024$ |
| | | C-LOT | $0.077 \pm 0.000$ | $0.000 \pm 0.000$ | $0.482 \pm 0.014$ | $0.064 \pm 0.000$ | $0.000 \pm 0.000$ | $2.391 \pm 0.005$ |
| | | SG-LOT | $0.077 \pm 0.000$ | $0.000 \pm 0.000$ | $0.462 \pm 0.005$ | $0.064 \pm 0.000$ | $0.000 \pm 0.000$ | $2.163 \pm 0.040$ |
| | | EROT | $-1.205 \pm 0.000$ | – | $3.042 \pm 0.143$ | $-1.343 \pm 0.000$ | – | $13.241 \pm 0.704$ |
| | | C-EROT | $-1.205 \pm 0.000$ | $0.000 \pm 0.000$ | $0.713 \pm 0.033$ | $-1.343 \pm 0.000$ | $0.000 \pm 0.000$ | $3.289 \pm 0.085$ |
| | | SG-EROT | $-1.205 \pm 0.000$ | $0.000 \pm 0.000$ | $0.690 \pm 0.025$ | $-1.343 \pm 0.000$ | $0.000 \pm 0.000$ | $3.146 \pm 0.005$ |
| | Random | LOT | $0.078 \pm 0.000$ | – | $3.171 \pm 0.424$ | $0.057 \pm 0.000$ | – | $15.171 \pm 1.651$ |
| | | SG-LOT | $0.078 \pm 0.000$ | $0.000 \pm 0.000$ | $1.743 \pm 0.083$ | $0.057 \pm 0.000$ | $0.000 \pm 0.000$ | $5.938 \pm 0.409$ |
| | | EROT | $-1.206 \pm 0.000$ | – | $3.091 \pm 0.029$ | $-1.346 \pm 0.000$ | – | $10.911 \pm 0.320$ |
| | | SG-EROT | $-1.206 \pm 0.000$ | $0.000 \pm 0.000$ | $1.796 \pm 0.083$ | $-1.346 \pm 0.000$ | $0.000 \pm 0.000$ | $6.326 \pm 0.251$ |
| $D_3$ | Block | LOT | $0.103 \pm 0.000$ | – | $3.332 \pm 0.440$ | $0.059 \pm 0.000$ | – | $22.090 \pm 1.805$ |
| | | C-LOT | $0.103 \pm 0.000$ | $0.000 \pm 0.000$ | $0.491 \pm 0.012$ | $0.059 \pm 0.000$ | $0.000 \pm 0.000$ | $2.313 \pm 0.072$ |
| | | SG-LOT | $0.103 \pm 0.000$ | $0.000 \pm 0.000$ | $0.208 \pm 0.007$ | $0.059 \pm 0.000$ | $0.000 \pm 0.000$ | $0.877 \pm 0.043$ |
| | | EROT | $-1.193 \pm 0.000$ | – | $3.046 \pm 0.169$ | $-1.339 \pm 0.000$ | – | $11.316 \pm 0.138$ |
| | | C-EROT | $-1.193 \pm 0.000$ | $0.000 \pm 0.000$ | $0.700 \pm 0.022$ | $-1.339 \pm 0.000$ | $0.000 \pm 0.000$ | $3.093 \pm 0.032$ |
| | | SG-EROT | $-1.193 \pm 0.000$ | $0.000 \pm 0.000$ | $0.335 \pm 0.004$ | $-1.339 \pm 0.000$ | $0.000 \pm 0.000$ | $1.421 \pm 0.007$ |
| | Random | LOT | $0.081 \pm 0.000$ | – | $2.997 \pm 0.426$ | $0.073 \pm 0.000$ | – | $15.911 \pm 1.651$ |
| | | SG-LOT | $0.081 \pm 0.000$ | $0.000 \pm 0.000$ | $1.618 \pm 0.016$ | $0.073 \pm 0.000$ | $0.000 \pm 0.000$ | $4.476 \pm 0.125$ |
| | | EROT | $-1.202 \pm 0.000$ | – | $3.209 \pm 0.129$ | $-1.336 \pm 0.000$ | – | $12.903 \pm 0.019$ |
| | | SG-EROT | $-1.202 \pm 0.000$ | $0.000 \pm 0.000$ | $1.623 \pm 0.089$ | $-1.336 \pm 0.000$ | $0.000 \pm 0.000$ | $4.730 \pm 0.427$ |
| $T$ | Block | LOT | $0.228 \pm 0.000$ | – | $3.581 \pm 0.378$ | $0.165 \pm 0.000$ | – | $18.551 \pm 1.398$ |
| | | C-LOT | $0.228 \pm 0.000$ | $0.000 \pm 0.000$ | $0.508 \pm 0.031$ | $0.165 \pm 0.000$ | $0.000 \pm 0.000$ | $2.533 \pm 0.099$ |
| | | SG-LOT | $0.228 \pm 0.000$ | $0.000 \pm 0.000$ | $0.157 \pm 0.017$ | $0.165 \pm 0.000$ | $0.000 \pm 0.000$ | $0.557 \pm 0.064$ |
| | | EROT | $-0.943 \pm 0.000$ | – | $2.928 \pm 0.015$ | $-1.104 \pm 0.000$ | – | $11.526 \pm 0.475$ |
| | | C-EROT | $-0.943 \pm 0.000$ | $0.000 \pm 0.000$ | $0.663 \pm 0.014$ | $-1.104 \pm 0.000$ | $0.000 \pm 0.000$ | $2.945 \pm 0.056$ |
| | | SG-EROT | $-0.943 \pm 0.000$ | $0.000 \pm 0.000$ | $0.212 \pm 0.036$ | $-1.104 \pm 0.000$ | $0.000 \pm 0.000$ | $0.844 \pm 0.001$ |
| | Random | LOT | $0.209 \pm 0.000$ | – | $3.595 \pm 0.418$ | $0.173 \pm 0.000$ | – | $18.295 \pm 2.133$ |
| | | SG-LOT | $0.209 \pm 0.000$ | $0.000 \pm 0.000$ | $1.493 \pm 0.086$ | $0.173 \pm 0.000$ | $0.000 \pm 0.000$ | $4.315 \pm 0.268$ |
| | | EROT | $-0.954 \pm 0.000$ | – | $2.942 \pm 0.066$ | $-1.104 \pm 0.000$ | – | $11.744 \pm 0.374$ |
| | | SG-EROT | $-0.954 \pm 0.000$ | $0.000 \pm 0.000$ | $1.553 \pm 0.019$ | $-1.104 \pm 0.000$ | $0.000 \pm 0.000$ | $4.182 \pm 0.358$ |
| $I$ | Block | LOT | $0.218 \pm 0.000$ | – | $3.377 \pm 0.312$ | $0.200 \pm 0.000$ | – | $16.718 \pm 1.481$ |
| | | C-LOT | $0.218 \pm 0.000$ | $0.000 \pm 0.000$ | $0.259 \pm 0.018$ | $0.200 \pm 0.000$ | $0.000 \pm 0.000$ | $1.267 \pm 0.149$ |
| | | SG-LOT | $0.218 \pm 0.000$ | $0.000 \pm 0.000$ | $0.112 \pm 0.007$ | $0.200 \pm 0.000$ | $0.000 \pm 0.000$ | $0.412 \pm 0.030$ |
| | | EROT | $-0.923 \pm 0.000$ | – | $2.766 \pm 0.119$ | $-1.063 \pm 0.000$ | – | $10.970 \pm 0.397$ |
| | | C-EROT | $-0.923 \pm 0.000$ | $0.000 \pm 0.000$ | $0.377 \pm 0.009$ | $-1.063 \pm 0.000$ | $0.000 \pm 0.000$ | $1.678 \pm 0.067$ |
| | | SG-EROT | $-0.923 \pm 0.000$ | $0.000 \pm 0.000$ | $0.128 \pm 0.006$ | $-1.063 \pm 0.000$ | $0.000 \pm 0.000$ | $0.475 \pm 0.032$ |
| | Random | LOT | $0.261 \pm 0.000$ | – | $3.592 \pm 0.480$ | $0.226 \pm 0.000$ | – | $19.004 \pm 0.082$ |
| | | SG-LOT | $0.261 \pm 0.000$ | $0.000 \pm 0.000$ | $1.517 \pm 0.045$ | $0.226 \pm 0.000$ | $0.000 \pm 0.000$ | $4.016 \pm 0.176$ |
| | | EROT | $-0.911 \pm 0.000$ | – | $2.548 \pm 0.006$ | $-1.044 \pm 0.000$ | – | $11.449 \pm 0.199$ |
| | | SG-EROT | $-0.911 \pm 0.000$ | $0.000 \pm 0.000$ | $1.378 \pm 0.016$ | $-1.044 \pm 0.000$ | $0.000 \pm 0.000$ | $4.035 \pm 0.265$ |

