# OpenReview forum: "Optimal Transport with Symmetry Groups"
_ICML.cc/2026/Conference — ICML 2026 regular_

### Official Review · Reviewer_Mv35 · 2026-03-10

**Soundness:** 4
**Presentation:** 4
**Significance:** 4
**Originality:** 4
**Overall Recommendation:** 5
**Confidence:** 2

**Summary:**

This paper proposes a new discrete optimal transport framework that generalizes traditional LOT, ROT, and EROT by incorporating symmetry for arbitrary finite groups. It also introduces a novel method to automatically identify the orbit structure directly from the cost matrix. Analytical results demonstrate that reducing the number of variables to the orbit space dramatically drops the computation cost. Experimental results further validate that the efficiency of the OR-SG-based framework provides a significant advantage over existing state-of-the-art methods.

**Compliance With Llm Reviewing Policy:**

Affirmed.

**Key Questions For Authors:**

N/A

**Strengths And Weaknesses:**

The paper demonstrates strong theoretical soundness. It provides rigorous mathematical proofs for all core Lemmas and Propositions, particularly in establishing the group-invariant property of the optimal transport plan. A key strength is the treatment of Assumption 3.3; the authors not only prove its generic validity using Lebesgue measure arguments but also proactively present a pathological counter-example in Appendix A.5 to define the algorithm's boundary. The empirical evaluation on both synthetic data and the NYU Symmetric Image Dataset consistently validates the theoretical speedup claims.
The presentation is logical and highly structured. While the mathematical density is high, but with proper prior knowledge is possible to follow up.
This work holds significant value for the Optimal Transport community. By reducing the computational complexity from the total number of points to the number of orbits, it addresses a major bottleneck in large-scale OT applications. As OT is a fundamental tool in machine learning and data analysis, providing a framework that speedup while maintaining exact optimality is a substantial contribution.
The originality lies in the considering discrete OT with non-cycling group and automatic symmetry-group identification to lower the computation cost.

---

> ### Author Rebuttal · Authors · 2026-03-29
>
> We thank the reviewer for the careful reading and positive feedback. We are glad that the reviewer finds our analysis, in particular the zero-measure set argument and the clarification via counterexamples, convincing. We will further improve the presentation and release the code upon acceptance.

---

> > ### Author Rebuttal · Reviewer_Mv35 · 2026-04-03
> >
> > Thank you for the authors' response.

---

### Official Review · Reviewer_N1X3 · 2026-03-11

**Soundness:** 3
**Presentation:** 2
**Significance:** 3
**Originality:** 3
**Overall Recommendation:** 4
**Confidence:** 5

**Summary:**

The work studies the discrete optimal transport problem where the input measures possess a symmetry group, and proposes a fast and efficient method for solving it.

**Compliance With Llm Reviewing Policy:**

Affirmed.

**Final Justification:**

My concerns regarding runtime and memory have been satisfactorily addressed by the additional experimental details. With respect to writing clarity, I trust the authors will further improve the manuscript in the revision. I have therefore increased my overall score.

**Key Questions For Authors:**

- The authors could consider providing additional experiments addressing the concerns on memory usage and ablation studies mentioned in **Weaknesses**. This point is important for me when considering whether to increase my score for this work.

- Given the current state of the manuscript, the authors should carefully review and revise the writing for clarity and correctness. A more thorough editing pass would help improve readability and reduce potential confusion for readers.

**Limitations:**

yes

**Strengths And Weaknesses:**

**Strengths**

- The problem addressed in this work is well-motivated and novel. It extends the previous framework developed for cyclic symmetry groups to more general finite symmetry groups.

- The theoretical results are supported by appropriate proofs. The assumptions adopted in the paper are clearly stated and well motivated. The authors also provide reasonable explanations for these assumptions, and the experimental results appear solid.

**Weaknesses**

- The overall contribution appears somewhat limited and largely builds upon previous work. In particular, the framework seems to follow relatively directly from earlier studies on cyclic symmetry groups. However, I leave the final judgment on the significance of this extension to other reviewers, and this concern does not currently affect my evaluation.

- The writing requires improvement in clarity. There are several errors and instances of awkward phrasing that may confuse readers who are not familiar with group-theoretic concepts. For example: (1) the definition of the stabilizer (lines 84–85) appears to be incorrect; (2) the presentation of the SG-ROT problem could be improved. One possible structure would be to first introduce Assumption 3.1, then present the formulation in Eq. (6), and finally explain why this formulation naturally leads to the SG-ROT problem. In the current presentation, the logical order feels somewhat reversed, which created confusion during my first reading. (To be clear, this does not affect my evaluation; it is simply a suggestion for improving clarity.)

- The symmetry groups and the dimension of Euclidean spaces considered in the experiments are relatively small. Moreover, the paper does not report the memory usage of the proposed method. In optimal transport methods, comparing memory consumption across methods is often an important aspect of evaluation.

- Although the authors provide theoretical runtime complexity, the experiments in the paper appear to be relatively small-scale and somewhat toy-like. It would be useful to include empirical validation of the claimed complexity. For instance, the authors could provide plots showing runtime and memory usage as functions of the number of points in the input measures, the size of the symmetry group $G$, or related parameters.

- No code has been provided with the submission.

---

> ### Author Rebuttal · Authors · 2026-03-29
>
> We sincerely thank you for your careful and constructive review. We have addressed the concerns and added additional experiments and clarifications as detailed below.
>
> ## Response to Question 1: More Detailed Experimental Reporting
>
> We agree that runtime and memory usage are both important. To address this, we run additional experiments with group $D_3$ and scale the number of points from 4800 to 14400.
>
> **Table 1: Runtime comparison (s).**
>
> | Ordering | Method  | 4800        | 7200         | 9600         | 12000        | 14400        |
> | - | - | - | - | - | - | - |
> | Block    | LOT     | 2.58 ± 0.21 | 11.35 ± 0.85 | 17.58 ± 0.24 | 34.64 ± 2.18 | 37.11 ± 0.80 |
> | Block    | C-LOT   | 0.39 ± 0.00 | 1.02 ± 0.00  | 1.91 ± 0.00  | 3.12 ± 0.00  | 4.70 ± 0.02  |
> | Block    | SG-LOT  | 0.16 ± 0.00 | 0.38 ± 0.01  | 0.71 ± 0.00  | 1.12 ± 0.01  | 1.79 ± 0.01  |
> | Block    | EROT    | 2.66 ± 0.01 | 7.02 ± 0.64  | 10.59 ± 0.26 | 18.08 ± 0.09 | 23.46 ± 0.72 |
> | Block    | C-EROT  | 0.53 ± 0.00 | 1.40 ± 0.01  | 2.63 ± 0.01  | 4.19 ± 0.01  | 6.62 ± 0.00  |
> | Block    | SG-EROT | 0.27 ± 0.00 | 0.59 ± 0.01  | 1.08 ± 0.00  | 1.82 ± 0.01  | 2.90 ± 0.02  |
> | Random   | LOT     | 2.68 ± 0.19 | 6.04 ± 0.47  | 12.28 ± 0.28 | 20.43 ± 1.45 | 33.10 ± 0.79 |
> | Random   | SG-LOT  | 1.63 ± 0.08 | 2.80 ± 0.11  | 4.41 ± 0.14  | 6.70 ± 0.06  | 17.04 ± 0.10 |
> | Random   | EROT    | 2.64 ± 0.00 | 5.75 ± 0.02  | 9.24 ± 0.02  | 16.21 ± 0.32 | 24.07 ± 0.41 |
> | Random   | SG-EROT | 1.53 ± 0.05 | 2.59 ± 0.02  | 4.12 ± 0.01  | 6.43 ± 0.01  | 16.88 ± 0.02 |
>
> **Table 2: Memory comparison (CPU/GPU) (MB).**
>
> | Ordering | Method  | 4800                | 7200                   | 9600                  | 12000                 | 14400                 |
> | - | - | - | - | - |- | - |
> | Block    | LOT     | 811.6 ± 86.9        | 1828.4 ± 196.8         | 3594.1 ± 8.2          | 5631.7 ± 0.1          | 7641.0 ± 471.1        |
> | Block    | C-LOT   | 446.5 ± 16.4        | 985.7 ± 12.1           | 1822.3 ± 6.5          | 2869.9 ± 22.2         | 4179.1 ± 22.2         |
> | Block    | SG-LOT  | 389.7 ± 11.3        | 912.0 ± 9.7            | 1598.7 ± 6.0          | 2500.8 ± 16.3         | 3646.3 ± 4.1          |
> | Block    | EROT    | 718.4 ± 1.9         | 1624.6 ± 0.8           | 2894.9 ± 5.9          | 4518.2 ± 5.9          | 6513.4 ± 12.4         |
> | Block    | C-EROT  | 467.1 ± 8.3         | 1061.1 ± 4.0           | 1898.2 ± 0.4          | 2970.5 ± 2.9          | 4277.6 ± 4.0          |
> | Block    | SG-EROT | 352.3 ± 6.1         | 880.5 ± 1.9            | 1622.2 ± 14.0         | 2544.1 ± 12.3         | 3643.0 ± 0.2          |
> | Random   | LOT     | 809.0 ± 88.0        | 1827.1 ± 200.8         | 3251.7 ± 350.1        | 5612.6 ± 16.4         | 8045.4 ± 64.9         |
> | Random   | SG-LOT  | 709.4 ± 2.3 / 483.4 | 1109.5 ± 11.2 / 1025.1 | 1857.2 ± 0.2 / 1847.2 | 2727.4 ± 3.4 / 2862.6 | 4756.4 ± 9.4 / 4099.5 |
> | Random   | EROT    | 719.1 ± 5.1         | 1618.9 ± 13.0          | 2891.0 ± 1.7          | 4523.1 ± 6.7          | 6517.4 ± 5.6          |
> | Random   | SG-EROT | 307.9 ± 3.4 / 483.4 | 735.9 ± 1.5 / 1025.1   | 958.1 ± 3.4 / 1847.2  | 1494.8 ± 9.0 / 2862.6 | 2155.8 ± 6.9 / 4099.5 |
>
> With block ordering, our method achieves the fastest runtime and lowest memory. Under random ordering, despite additional steps, runtime remains significantly lower than standard OT, with comparable memory shown as “CPU / GPU”.
>
> We also report a runtime breakdown. The cost can be computed by scaling the OR-SG-ROT solution by $|G|$. We report this step separately under the “Cost” column.
>
> **Table 3: Runtime breakdown (s).**
>
> | Method  | Points | Total        | Permute      | Solve       | Cost        | Reconstruct |
> | - | - | - | - | - | - | - |
> | SG-LOT  | 4800   | 1.63 ± 0.08  | 0.78 ± 0.09  | 0.09 ± 0.00 | 0.02 ± 0.00 | 0.74 ± 0.00 |
> | SG-EROT | 4800   | 1.53 ± 0.05  | 0.66 ± 0.04  | 0.10 ± 0.00 | 0.08 ± 0.00 | 0.69 ± 0.01 |
> | SG-LOT  | 7200   | 2.80 ± 0.11  | 1.17 ± 0.14  | 0.20 ± 0.00 | 0.05 ± 0.00 | 1.37 ± 0.03 |
> | SG-EROT | 7200   | 2.59 ± 0.02  | 0.90 ± 0.01  | 0.18 ± 0.00 | 0.17 ± 0.00 | 1.33 ± 0.01 |
> | SG-LOT  | 9600   | 4.41 ± 0.14  | 1.71 ± 0.03  | 0.36 ± 0.00 | 0.09 ± 0.00 | 2.25 ± 0.11 |
> | SG-EROT | 9600   | 4.12 ± 0.01  | 1.31 ± 0.04  | 0.35 ± 0.00 | 0.32 ± 0.02 | 2.14 ± 0.01 |
> | SG-LOT  | 12000  | 6.70 ± 0.06  | 2.31 ± 0.08  | 0.63 ± 0.01 | 0.18 ± 0.00 | 3.59 ± 0.01 |
> | SG-EROT | 12000  | 6.43 ± 0.01  | 1.87 ± 0.06  | 0.60 ± 0.00 | 0.50 ± 0.01 | 3.45 ± 0.06 |
> | SG-LOT  | 14400  | 17.04 ± 0.10 | 10.99 ± 0.05 | 0.96 ± 0.02 | 0.26 ± 0.01 | 4.83 ± 0.02 |
> | SG-EROT | 14400  | 16.88 ± 0.02 | 10.47 ± 0.04 | 0.87 ± 0.01 | 0.83 ± 0.00 | 4.70 ± 0.06 |
>
> Code will be released upon acceptance.
>
> ## Response to Question 2: Writing Revisions
>
> We thank the reviewer for pointing out the typo. The stabilizer should be
> $$
> \mathrm{Stab}(x) := \lbrace g \in G^E \mid gx = x \rbrace.
> $$
> We also agree with the suggested reordering and will adjust the presentation accordingly to improve clarity.

---

> > ### Author Rebuttal · Reviewer_N1X3 · 2026-04-02
> >
> > Thank you for the response.
> >
> > My concerns regarding runtime and memory have been satisfactorily addressed by the additional experimental details. With respect to writing clarity, I trust the authors will further improve the manuscript in the revision. I have therefore increased my overall score.

---

> > > ### Author Response · Authors · 2026-04-02
> > >
> > > Thank you for your recognition and support. We are glad our additional experiments resolved your concerns, and your suggestion on writing clarity will be carefully incorporated. Many thanks for your constructive feedback.

---

### Official Review · Reviewer_yfQN · 2026-03-12

**Soundness:** 3
**Presentation:** 3
**Significance:** 2
**Originality:** 3
**Overall Recommendation:** 4
**Confidence:** 5

**Summary:**

The authors propose a novel algorithm for optimal transport which exploits intrinsic symmetries induced by group action in the transport cost to reduce the optimal transport alignment to a simplified, reduced problem. The authors formulate this problem by introducing permutations $T_g^s$ and $T_g^t$ for which the cost C is invariant under the action of G and for which $ T_g^s C (T_g^t)^\top = C$ and $ T_g^s P (T_g^t)^\top = P$ for $g\in G$. The authors formulate a full symmetry-constrained problem SG-ROT and a reduced problem on the orbit space OR-SG-ROT. This proceeds by using the cost matrix to identify which points belong in common orbits, a condition which applies when points in distinct orbits yield distinct distance patterns in the rows/columns of the cost matrix. In practice, the authors permute the rows of the cost into a reducible ordering, solve OR-SG-ROT, and reconstruct the reducible solution to yield the solution to the full problem. The authors demonstrate the advantages of leveraging symmetry empirically on a number of real and synthetic datasets with varying degrees of symmetry.

**Compliance With Llm Reviewing Policy:**

Affirmed.

**Final Justification:**

Overall, incorporating symmetry groups into OT is interesting and natural in certain settings. Some reviewers (e.g., frgj) did note that the  result is perhaps somewhat straightforward/immediate, and I also do think the utility may be restricted. That said, the work is technically sound and well written, so I lean toward accept.

**Key Questions For Authors:**

- Perhaps the most interesting question to me is whether the authors can comment on the relation to Noether’s theorem (i.e., that every symmetry has an associated conservation law), and whether the formulation of OT with symmetries can also be viewed from a perspective of OT with conserved quantities along the Wasserstein geodesic trajectories?
- It would be valuable to mention the line of work using low-rank structure in the coupling matrix as well (the field of low-rank OT, e.g. Scetbon and Cuturi ‘21, ‘22, Forrow ‘19, Halmos ‘25). These have parallels with the orbit-space reduction of SG-OT and the noted approaches tackling structure in the cost/data (Tenetov ‘18, Altschuler ‘19, Takeda ‘24, Takeda ‘25) from the complementary perspective of the OT coupling.
- Can the authors comment on how significant symmetries (or approximate symmetries) are in more practical, unstructured data such as image distributions?

**Limitations:**

Yes

**Strengths And Weaknesses:**

Strengths:

- Overall, the technique is well-justified, and there are many problems where one might seek to apply optimal transport with known symmetry groups. In this case, the symmetry-based reduction of the OT problem to the orbit space is elegant and offers significant computational advantages which the authors have demonstrated convincingly.

- In the case that one knows the symmetry groups of their data a priori, as might be the case in chemical or physical data of e.g. molecules with $C_{n}, S_{n}, D_{n}, T_{n}$ etc., this approach is natural and immediately offers optimal solutions with significant acceleration.

- The authors provide significant background regarding approaches for OT which similarly use low-rankness or structure in the input data and cost (Tenetov ‘18, Altschuler ‘19, Takeda ‘24, Takeda ‘25, and so on). Relative to the body of work which leverages structure in the cost, the approach of using symmetry groups and reducing to the quotient space is novel and interesting.

Weaknesses:

- Using distance profiles for identifying symmetry groups is, as noted by the authors, a heuristic. Sorting rows is relatively cheap, but having matching patterns is only necessary and not sufficient in general as a condition to be in a shared orbit. Actually finding the automorphism group of the cost C satisfying the $ T_g^s C (T_g^t)^\top = C$ condition is quasipolynomial and (practically) exponential. So it’s acceptable for rejecting non-orbits, but the procedure can’t exactly identify exact orbits. Since OT is already a convex, poly-time problem, unless the data scale is massive with very significant symmetries the approach may return an approximate solution without offering much of a practical advantage in complexity in return.

- For a matrix of size $n \times n$ the sorting needed for orbit identification/permutation is $n^2 \log n$. This is the same as the iteration complexity of the Sinkhorn algorithm (Cuturi ‘13) already, so that the overhead from orbit identification is still notable for an approximate solution, which the entropy-regularized Sinkhorn algorithm can provide.

References:

Jason Altschuler and Francis Bach and Alessandro Rudi and Jonathan Niles-Weed, Massively scalable Sinkhorn distances via the Nystr\"om method, NeurIPS 2019

Takeda, S., Akagi, Y., Marumo, N., & Niwa, K. (2024). Optimal Transport with Cyclic Symmetry. Proceedings of the AAAI Conference on Artificial Intelligence, 38(14), 15211-15221

Shoichiro Takeda, Yasunori Akagi; Gromov-Wasserstein Problem with Cyclic Symmetry, Proceedings of the IEEE/CVF Conference on Computer Vision and Pattern Recognition (CVPR), 2025, pp. 21011-21020

Tenetov, Evgeny and Wolansky, Gershon and Kimmel, Ron, Fast Entropic Regularized Optimal Transport Using Semidiscrete Cost Approximation, SIAM Journal on Scientific Computing 2018

Cuturi, Marco, Sinkhorn Distances: Lightspeed Computation of Optimal Transport, NeurIPS 2013

Scetbon, Meyer and Cuturi, Marco and Peyr{\'e}, Gabriel, Low-Rank Sinkhorn Factorization, Proceedings of the 38th International Conference on Machine Learning 2021

Scetbon, Meyer and Cuturi, Marco, Low-rank Optimal Transport: Approximation, Statistics and Debiasing, 36th Conference on Neural Information Processing Systems (NeurIPS 2022)

Peter Halmos and Julian Gold and Xinhao Liu and Benjamin Raphael, Hierarchical Refinement: Optimal Transport to Infinity and Beyond, Forty-second International Conference on Machine Learning 2025

Aden Forrow, Jan-Christian Hütter, Mor Nitzan, Philippe Rigollet, Geoffrey Schiebinger, Jonathan Weed, Statistical Optimal Transport via Factored Couplings, Proceedings of the Twenty-Second International Conference on Artificial Intelligence and Statistics

---

> ### Author Rebuttal · Authors · 2026-03-29
>
> ## Response to Question 1: Connection between Noether’s theorem and SG-ROT
>
> We thank the reviewer for this inspiring perspective. Noether’s theorem reveals a deep connection between continuous symmetries and conservation laws in dynamical systems: every continuous symmetry of the Lagrangian corresponds to a conserved quantity. While this is a powerful result in the continuous domain, our work focuses on discrete optimal transport problems. Nevertheless, the principle of symmetry reduction extends naturally to the continuous setting.
>
> Let $G$ be a compact group acting on spaces $ X, Y $ with cost $ c(g \cdot x, g \cdot y) = c(x, y) $ and measures $g_{\sharp}\mu = \mu$, $g_{\sharp}\nu = \nu$. We show that a $G$-invariant optimal coupling exists. Let $h_G$ be the normalized Haar measure on $G$. Take any optimal coupling $\pi$ and define
>
> $$
> \pi_G = \int_G (g,g)_{\sharp} \pi \ \mathrm{d}h_G(g),
> $$
>
> where $(g,g)_{\sharp}\pi$ is the pushforward under $ (x,y) \mapsto (g \cdot x, g \cdot y) $. Then:
>
> - **Marginals:** For measurable $A \subseteq X$,
>
>   $$
>   \pi_G(A \times Y)
>   = \int_G \pi(g^{-1}A \times Y)\ \mathrm{d}h_G(g)
>   = \int_G \mu(g^{-1}A)\ \mathrm{d}h_G(g)
>   = \mu(A).
>   $$
>
>   Similarly for the second marginal. Hence $ \pi_G \in \Pi(\mu,\nu) $.
>
> - **Invariance.** By the left-invariance of the Haar measure, for any $h\in G$ we have
>   $
>   (h,h)_{\sharp}\pi_G
>   = \pi_G.
>   $
>
> - **Optimality:** By cost invariance,
>
>   $$
>   \int c\ \mathrm{d}\pi_G
>   = \int_G \int c(g \cdot x, g \cdot y)\ \mathrm{d}\pi\ \mathrm{d}h_G(g)
>   = \int c\ \mathrm{d}\pi.
>   $$
>
> Thus $\pi_G$ is a $G$-invariant optimal coupling. This continuous analogue of Lemma 3.2 will be discussed in the final version.
>
> ## Response to Question 2: Discussion of Low-Rank OT
>
> We thank the reviewer for this suggestion. Low-rank optimal transport [1–3] exploits low-rank structure in the coupling matrix for computational gains. Our work takes a complementary approach, leveraging geometric symmetry in the cost matrix and measures to reduce the problem to the orbit space. These two structural assumptions are complementary and not mutually exclusive. We will add references and clarify this distinction in the Introduction and Discussion; exploring their combination is a promising future direction.
>
> ## Response to Question 3: Prevalence of Symmetry in Unstructured Data
>
> We agree that exact geometric symmetry is indeed less common in fully unstructured data such as natural images. However, we believe our method remains valuable in two important scenarios, which we now substantiate with concrete datasets and references.
>
> **1. Strictly symmetric data.**
> Our method is particularly suited to domains where symmetry is inherent and can be explicitly modeled. For example:
>
> - **Molecular and crystalline structures:** chemical molecules and crystal structures can be naturally modeled as graphs, which often exhibit rich symmetry. Such structural symmetries are fundamental in chemical graph theory [4].
> - **Synthetic point clouds:** as demonstrated in our experiments, we can generate point clouds with prescribed symmetry groups (cyclic, dihedral, tetrahedral, etc.) to benchmark and validate algorithmic performance under strict symmetry.
>
> **2. Approximately symmetric data.**
> Even when symmetry is only approximate, our approach remains effective with controlled degradation. In our paper, we validated this on:
>
> - **NYU Symmetric Image Dataset** [5]: images only approximately satisfy the symmetry condition due to natural variations in pixel intensities, providing a testbed for measure approximation.
> - **ShapeNet** [6]: objects with inherent geometric symmetry can be pre-processed to yield strictly symmetric point clouds, as done in previous work [7].
>
> **References**
>
> [1] Scetbon, M., & Cuturi, M. Low-rank optimal transport: Approximation, statistics and debiasing. In *Advances in Neural Information Processing Systems*, 2021.
>
> [2] Forrow, A., Hütter, J. C., Nitzan, M., Rigollet, P., Schiebinger, G., & Weed, J. Statistical optimal transport via factored couplings. In *International Conference on Artificial Intelligence and Statistics*, 2019.
>
> [3] Halmos, P., Gold, J., Liu, X., & Raphael, B. Hierarchical refinement: Optimal transport to infinity and beyond. In *International Conference on Machine Learning*, 2025.
>
> [4] Bonchev, D. Chemical graph theory: introduction and fundamentals, volume 1. CRC Press, 1991.
>
> [5] Cicconet, M., Birodkar, V., Lund, M., Werman, M., & Geiger, D. A convolutional approach to reflection symmetry. *Pattern Recognition Letters*, 95, 44–50, 2017.
>
> [6] Chang, A. X., Funkhouser, T., Guibas, L., et al. ShapeNet: An Information-Rich 3D Model Repository. *arXiv:1512.03012*, 2015.
>
> [7] Takeda, S., & Akagi, Y. Gromov-Wasserstein problem with cyclic symmetry. In *Proceedings of the IEEE/CVF Conference on Computer Vision and Pattern Recognition (CVPR)*, pages 21011–21020, 2025.

---

> > ### Author Rebuttal · Reviewer_yfQN · 2026-03-31
> >
> > Thank you, my questions have been resolved. I maintain my score, which leans towards accept.

---

> > > ### Author Response · Authors · 2026-04-02
> > >
> > > Thank you again for your inspiring and constructive feedback. It has helped us better articulate the theoretical depth and positioning of our work. We are glad that our responses have fully addressed your concerns.

---

### Official Review · Reviewer_frgj · 2026-03-22

**Soundness:** 3
**Presentation:** 3
**Significance:** 1
**Originality:** 2
**Overall Recommendation:** 3
**Confidence:** 4

**Summary:**

This paper studies reduced formulations of optimal transport when the source and target measures satisfy symmetries of the original metric space. The main contribution of the paper is to reduce the computation to the total number of points (of each measure) divided by a factor Cardinal(G) where G is the isometry group, which acts freely on the space. This reduction is obtained after first sorting the cost matrix along rows and columns.
Experiments are shown on symmetric data, but also on approximate symmetric data.

**Compliance With Llm Reviewing Policy:**

Affirmed.

**Final Justification:**

The content of the paper is correct and interesting, yet I am not convinced of its potential impact.

**Key Questions For Authors:**

Why shall we care about optimal transport with symmetries: are there contexts in which this reduction would play a crucial role?

What about possible approximations of the cost matrix or the space, source, and target, with symmetries, in the direction of section 5.2 and 5.3 ?

**Limitations:**

yes.

**Strengths And Weaknesses:**

The paper is technically sound, and its theoretical analysis is correct. The paper is also well written.
One interesting contribution is to detect symmetries from the cost matrix itself, by the collections of distances. This requires some assumptions, but does not necessitate the explicit definition of symmetries.

My major concern is the following: This result is a low-hanging fruit in itself, that is this result is very much expected. This, in itself, would not be an issue, if the significance or practical applications were important. However, I did not see very convincing set up in which such an approach could lead to a dramatic improvement.
Most of the data do not present such interesting symmetries, however, it is possible that in some context (e.g. molecules or else) this reductions is key.
If the authors can give more compelling arguments for why this reduction is important, I would be happy to raise my score.

---

> ### Author Rebuttal · Authors · 2026-03-29
>
> ## Response to Question 1: Motivation and Practical Application Scenarios
>
> We thank the reviewer for the questions on the motivation and application scenarios of our work.
>
> Symmetry is prevalent in real-world objects, and optimal transport (OT) maps between such objects often naturally exhibit symmetry invariance. This makes symmetry reduction a particularly natural and appealing approach. While our paper focuses on general finite groups, the key takeaway is the broad applicability of our algorithm across diverse symmetry structures—even simple symmetries can already yield multiplicative speedups.
>
> To further demonstrate its practical potential, we present two additional experiments on ShapeNet [1].
>
> **1. Axially symmetric data (car).**
>  We select six axially symmetric models and compute OT between all pairs. In this setting all methods recover the same solution. As shown in the table below, both SG-LOT and SG-EROT achieve clear speedups.
>
> | Method  | Time (s)        |
> | ------- | --------------- |
> | LOT     | 1.1498 ± 0.0895 |
> | SG-LOT  | 0.3068 ± 0.0197 |
> | EROT    | 0.5011 ± 0.0252 |
> | SG-EROT | 0.2247 ± 0.0102 |
>
> **2. Rotationally symmetric data (lamp).**
>  For rotationally symmetric models, we first take a subset within the angular range $(0, 360/n)$, and then reconstruct the full point cloud by rotating it $n$ times, resulting in cyclic symmetry of order $n$. We evaluate OT between all model pairs for $n = 2, 4, 6, 12$. As shown below, the speedup becomes more pronounced as $n$ increases. For example, when $n = 12$, the runtime of SG-LOT and SG-EROT is only **2.6%** and **9.6%** of that of standard LOT and EROT.
>
> |         | $Z_2$           | $Z_4$           | $Z_6$           | $Z_{12}$        |
> | ------- | --------------- | --------------- | --------------- | --------------- |
> | LOT     | 0.6929 ± 0.2384 | 0.7580 ± 0.2700 | 0.7450 ± 0.2636 | 0.7736 ± 0.2977 |
> | SG-LOT  | 0.1934 ± 0.0656 | 0.0585 ± 0.0180 | 0.0347 ± 0.0121 | 0.0202 ± 0.0067 |
> | EROT    | 0.2909 ± 0.0738 | 0.2943 ± 0.0731 | 0.3025 ± 0.0789 | 0.3013 ± 0.0961 |
> | SG-EROT | 0.1234 ± 0.0294 | 0.0630 ± 0.0179 | 0.0463 ± 0.0150 | 0.0290 ± 0.0096 |
>
> In summary, the theoretical contribution of this work lies in systematically connecting group symmetry with optimal transport, establishing the group invariance property, and providing a precise characterization of orbit identification (including its generic validity and explicit counterexamples). Building on this, we design a symmetry-reduced algorithm that is robust to input ordering.
>
> On the practical side, symmetry is prevalent in many real-world datasets. Our experiments—on ShapeNet cars and lamps, the NYU symmetric image dataset, and synthetic data with various groups—demonstrate that the method achieves significant speedups while preserving solution quality.
>
> Moreover, the framework can be naturally extended to Gromov–Wasserstein problems, further broadening its applicability. In particular, when two metric spaces share the same symmetry group, the GW problem can also be reduced to the orbit space, as shown by Takeda and Akagi [2] for cyclic symmetry.
>
> We believe these results provide a general framework that is both theoretically grounded and practically valuable.
>
> ## Response to Question 2: Impact of Approximate Symmetry
>
> Our framework assumes symmetry in both point positions and measures. In practice, approximate symmetry in the measure does not affect the algorithm. As seen in Experiments 5.2 and 5.3, the error remains controlled. This is also expected from the stability of OT under perturbations of the measure. We also note that this approximate symmetry can still be useful in practice. Prior work [3] uses such approximate solutions to initialize C-ROT and recover exact solutions. Similarly, our framework can be used as a warm start, providing both acceleration and improved convergence toward the exact solution.
>
> Approximate symmetry in point positions is more challenging, as it affects orbit identification. Our current method does not explicitly handle this case. Extending the approach to support approximate orbit identification is part of our future work.
>
> **References**
>
> [1] Chang A X, Funkhouser T, Guibas L, et al. Shapenet: An information-rich 3d model repository[J]. arXiv preprint arXiv:1512.03012, 2015.
>
> [2] Shoichiro Takeda and Yasunori Akagi. Gromov-Wasserstein problem with cyclic symmetry. In *Proceedings of the IEEE/CVF Conference on Computer Vision and Pattern Recognition (CVPR)*, pages 21011–21020, 2025.
>
> [3] Takeda S, Akagi Y, Marumo N, et al. Optimal transport with cyclic symmetry[C]//Proceedings of the AAAI Conference on Artificial Intelligence. 2024, 38(14): 15211-15221.

---

> > ### Author Rebuttal · Reviewer_frgj · 2026-04-02
> >
> > I reckon that the contribution is sound and the performance gain is clear in view of the experiments. I think that this result is worth publishing.
> > However, I am not convinced by the practical significance of the method because I do not have evidence of datasets on which this reduction is crucial:
> > My concern is how many times this algorithm will be used? If only used rarely, what would be the impact (maybe important)?
> >
> > For these reasons, I do not raise my score.

---

> > > ### Author Response · Authors · 2026-04-08
> > >
> > > We thank the reviewer for acknowledging the soundness and value of our work. To address the reviewer’s concern about application scenarios, we further supplement the following content.
> > >
> > > Optimal transport has been widely used for molecular graph comparison [1–3]. Specifically, we can convert molecular structures into graphs, compute Weisfeiler‑Lehman hash features for each node, and then use Hamming distance to build the cost matrix between molecules, finally computing optimal transport to describe the structural differences between molecular graphs. Molecular graph data themselves often possess rich symmetry (chemical graph theory [4] provides a systematic discussion). For symmetric molecules (e.g., methylbenzene, hydroxybenzene), if the atoms are ordered according to the block ordering in our paper, the cost matrix exhibits a block‑circulant structure at the order of the group size, and then using SG‑EROT instead of standard EROT achieves lossless acceleration.
> > >
> > > To verify this, we selected eight highly symmetric compounds: hexamethylbenzene, hexahydroxybenzene, hexamethoxybenzene, hexaaminobenzene, hexaformylbenzene, hexaacetylbenzene, hexacarboxybenzene, and hexabenzylbenzene. For all pairwise combinations (28 pairs) of these molecules, we performed molecular graph comparison using standard EROT and SG‑EROT respectively, and recorded the objective value, total runtime, and the distance between the transport plans obtained by the two methods. The average results are as follows:
> > >
> > > | Method  | Obj.value (mean ± std) | Time (mean ± std) (ms) | Solution distance (mean ± std) |
> > > | :------ | :--------------------- | :--------------------- | :----------------------------- |
> > > | EROT    | -3.045983 ± 0.000161   | 0.5850 ± 0.1590        | —                              |
> > > | SG‑EROT | -3.045982 ± 0.000161   | 0.2909 ± 0.0412        | 0.0000 ± 0.0000                |
> > >
> > > It can be seen that SG‑EROT maintains the same accuracy (the cost difference is on the order of 1e‑6, and the transport plan obtained by SG‑EROT is exactly the same as that of standard EROT), while the runtime is about half that of standard EROT. This verifies the effectiveness of our method on real molecular graph data.
> > >
> > > Furthermore, symmetry is also prevalent in 3D surface data such as faces and brain images. These are all active application areas of OT where symmetry is the norm rather than the exception. We hope these supplements help the reviewer better assess the practical value of our work.
> > >
> > > **References**
> > >
> > > [1] Nikolentzos, G., Meladianos, P., & Vazirgiannis, M. Matching Node Embeddings for Graph Similarity. *AAAI*, 2017.
> > >
> > > [2] Petric Maretic, H., et al. GOT: An Optimal Transport framework for Graph comparison. *NeurIPS*, 2019.
> > >
> > > [3] Togninalli, M.; Ghisu, E.; Llinares-López, F.; Rieck, B.; and Borgwardt, K. 2019. Wasserstein Weisfeiler–Lehman Graph Kernels. In *NeurIPS*, 6436–6446.
> > >
> > > [4] Bonchev, D. *Chemical graph theory: introduction and fundamentals*, Vol. 1. CRC Press, 1991.

---

### Decision · Program_Chairs · 2026-04-30

**Decision:**

Accept (regular)

**Comment:**

The authors propose Regularized Optimal Transport with Symmetry Groups (SG-ROT) for accelerating optimal transport (OT) by exploiting finite symmetry groups in the data. The main idea is to automatically detect orbit structure from the cost matrix, then reduce the OT problem to a smaller one on the orbit space. We think that the submission is technically sound, and theoretically well-founded. The proposed idea of leveraging symmetry for OT problem is elegant, which provides clear computational advantages for OT when symmetry is present. However, the reviewers also raised concerns on its restricted scope, i.e., its computational benefits rely on the presence of strong symmetry, which may be limited in general applications. Additionally, orbit identification is partly heuristic, less robust under approximate symmetry.

Overall, despite raised concerns on the restricted scope of its applicability, we think the submission provides a clear, well-developed, and practically useful contributions for problems involving presented symmetry. Its finding results are interesting for the community. Therefore, we recommend acceptance for the submission.